# Quantification of the pace of biological aging in humans through a blood test, the DunedinPoAm DNA methylation algorithm

Daniel W Belsky[1,2]*, Avshalom Caspi[3,4,5,6], Louise Arseneault[3], Andrea Baccarelli[7], David L Corcoran[6], Xu Gao[7], Eiliss Hannon[8], Hona Lee Harrington[4], Line JH Rasmussen[4,9], Renate Houts[4], Kim Huffman[10,11], William E Kraus[10,11], Dayoon Kwon[2], Jonathan Mill[8], Carl F Pieper[11,12], Joseph A Prinz[6], Richie Poulton[13], Joel Schwartz[14], Karen Sugden[4], Pantel Vokonas[15], Benjamin S Williams[4], Terrie E Moffitt[3,4,5,6]

[1]Department of Epidemiology, Columbia University Mailman School of Public Health, New York, United States; [2]Butler Columbia Aging Center, Columbia University Mailman School of Public Health, New York, United States; [3]Social, Genetic, and Developmental Psychiatry Centre, Institute of Psychiatry, King's College London, London, United Kingdom; [4]Department of Psychology and Neuroscience, Duke University, Durham, United States; [5]Department of Psychiatry and Behavioral Sciences, Duke University School of Medicine, Durham, United States; [6]Center for Genomic and Computational Biology, Duke University, Durham, United States; [7]Laboratory of Precision Environmental Health, Mailman School of Public Health, Columbia University, New York, United States; [8]University of Exeter Medical School, College of Medicine and Health, Exeter, United Kingdom; [9]Clinical Research Centre, Copenhagen University Hospital Amager and Hvidovre, Hvidovre, Denmark; [10]Duke Molecular Physiology Institute, Duke University, Durham, United States; [11]Duke University Center for the Study of Aging, Duke University, Durham, United States; [12]Department of Biostatistics, Duke University School of Medicine, Durham, United States; [13]Department of Psychology and Dunedin Multidisciplinary Health and Development Research Unit, University of Otago, Otago, New Zealand; [14]Department of Environmental Health Sciences, Harvard TH Chan School of Public Health, Boston, United States; [15]Veterans Affairs Normative Aging Study, Veterans Affairs Boston Healthcare System, Department of Medicine, Boston University School of Medicine, Boston, United States

*For correspondence:
db3275@cumc.columbia.edu

Competing interests: The authors declare that no competing interests exist.

**Abstract** Biological aging is the gradual, progressive decline in system integrity that occurs with advancing chronological age, causing morbidity and disability. Measurements of the pace of aging are needed as surrogate endpoints in trials of therapies designed to prevent disease by slowing biological aging. We report a blood-DNA-methylation measure that is sensitive to variation in pace of biological aging among individuals born the same year. We first modeled change-over-time in 18 biomarkers tracking organ-system integrity across 12 years of follow-up in n = 954 members of the Dunedin Study born in 1972–1973. Rates of change in each biomarker over ages 26–38 years were composited to form a measure of aging-related decline, termed Pace-of-Aging. Elastic-net regression was used to develop a DNA-methylation predictor of Pace-of-Aging, called DunedinPoAm for Dunedin(P)ace(o)f(A)ging(m)ethylation. Validation analysis in cohort studies and

the CALERIE trial provide proof-of-principle for DunedinPoAm as a single-time-point measure of a person's pace of biological aging.

## Introduction

Aging of the global population is producing forecasts of rising burden of disease and disability (*Harper, 2014*). Because this burden arises from multiple age-related diseases, treatments for single diseases will not address the burden challenge (*Goldman et al., 2013*). Geroscience research suggests an appealing alternative: treatments to slow aging itself could prevent or delay the multiple diseases that increase with advancing age, perhaps with a single therapeutic approach (*Gladyshev, 2016*; *Kaeberlein, 2013*). Aging can be understood as a gradual and progressive deterioration in biological system integrity (*Kirkwood, 2005*). This deterioration is thought to arise from an accumulation of cellular-level changes. These changes, in turn, increase vulnerability to diseases affecting many different organ systems (*Kennedy et al., 2014*; *López-Otín et al., 2013*). Animal studies suggest treatments that slow the accumulation of cellular-level changes can extend healthy lifespan (*Campisi et al., 2019*; *Kaeberlein et al., 2015*). However, human trials of these treatments are challenging because humans live much longer than model animals, making it time-consuming and costly to follow up human trial participants to test treatment effects on healthy lifespan. This challenge will be exacerbated in trials that will give treatments to young or middle-aged adults, with the aim to prevent the decline in system integrity that antedates disease onset by years. Involving young and midlife adults in healthspan-extension trials has been approved for development by the National Advisory Council on Aging (2019 CTAP report to NACA). In midlife trials of treatments to slow aging, called geroprotectors (*Moskalev et al., 2016*), traditional endpoints such as disease diagnosis or death are too far in the future to serve as outcomes. Translation of geroprotector treatments to humans could be aided by measures that quantify the pace of deterioration in biological system integrity in human aging. Such measures could be used as surrogate endpoints for healthy lifespan extension (*Justice et al., 2016*; *Justice et al., 2018*; *Moskalev et al., 2016*), even with young-to-midlife adult trial participants. A useful measure should be non-invasive, inexpensive, reliable, and highly sensitive to biological change.

Recent efforts to develop such measures have focused on blood DNA methylation as a biological substrate highly sensitive to changes in chronological age (*Fahy et al., 2019*; *Horvath and Raj, 2018*). Methylation-clock algorithms have been developed to identify methylation patterns that characterize individuals of different chronological ages. However, a limitation is that individuals born in different years have grown up under different historical conditions (*Schaie, 1967*). For example, people born 70 years ago experienced more exposure to childhood diseases, tobacco smoke, airborne lead, and less exposure to antibiotics and other medications, and lower quality nutrition, all of which leave signatures on DNA methylation (*Bell et al., 2019*). As a result, the clocks confound methylation patterns arising from early-life exposures to methylation-altering factors with methylation patterns related to biological aging during adulthood. An alternative approach is to study individuals who were all born the same year, and find methylation patterns that differentiate those who have been aging biologically faster or slower than their same-age peers. The current article reports four steps in our work toward developing a blood DNA methylation measure to represent individual variation in the pace of biological aging.

In Step 1, which we previously reported (*Belsky et al., 2015*), we collected a panel of 18 blood-chemistry and organ-system-function biomarkers at three successive waves of the Dunedin Study, which follows a 1972–73 population-representative one-year birth cohort (N = 1037). We used repeated-measures data collected when Study members were aged 26, 32, and 38 years old to quantify rates of biological change. We modelled the rate of change in each biomarker and calculated how each Study member's personal rate-of-change on that biomarker differed from the cohort norm. We then combined the 18 personal rates of change across the panel of biomarkers to compute a composite for each Study member that we called the Pace of Aging. Pace of Aging represents a personal rate of multi-organ-system decline over a dozen years. Pace of Aging was normally distributed, and showed marked variation among Study members who were all the same chronological age, confirming that individual differences in biological aging do emerge already by age 38, years before chronic disease onset.

**eLife digest** People's bodies age at different rates. Age-related biological changes that increase the risk of disease and disability progress rapidly in some people. In others, these processes occur at a slower pace, allowing those individuals to live longer, healthier lives. This observation has led scientists to try to develop therapies that slow aging. The hope is that such treatments could prevent or delay diseases like heart disease or dementia, for which older age is the leading risk factor.

Studies in animals have identified treatments that extend the creatures' lives and slow age-related disease. But testing these treatments in humans is challenging. Our lives are much longer than the worms, flies or mice used in the experiments. Scientists would have to follow human study participants for decades to detect delays in disease onset or an extension of their lives. An alternative approach is to try to develop a test that measures the pace of aging, or essentially "a speedometer for aging". This would allow scientists to more quickly determine if treatments slow the aging process.

Now, Belsky et al. show a blood test designed to measure the pace of aging predicts which people are at increased risk of poor health, chronic disease and an earlier death. First, data about chemical changes to an individual's DNA, called DNA methylation, were analyzed from white blood cell samples collected from 954 people in a long-term health study known as "The Dunedin Study". Using the data, Belsky et al. then developed an algorithm – named "DunedinPoAm" – that identified people with an accelerated or slowed pace of aging based on a single blood test. Next, they used the algorithm on samples from participants in three other long-term studies. This verified that those people the algorithm identified as aging faster had a greater risk of poor health, developing chronic diseases or dying earlier. Similarly, those identified as aging more slowly performed better on tests of balance, strength, walking speed and mental ability, and they also looked younger to trained raters. Additionally, Belsky et al. used the test on participants in a randomized trial testing whether restricting calories had potential to extend healthy lifespan. The results suggested that the calorie restriction could counter the effects of an accelerated pace of aging.

The test developed by Belsky et al. may provide an alternate way of measuring whether age-slowing treatments work. This would allow faster testing of treatments that can extend the healthy lifespan of humans. The test may also help identify individuals with accelerated aging. This might help public health officials test whether policies or programs can help people lead longer, healthier lives.

In Step 2, which we previously reported, we validated the Pace of Aging against known criteria. As compared to other Study members who were the same chronological age but had slower Pace of Aging, Study members with faster Pace of Aging performed more poorly on tests of physical function; showed signs of cognitive decline on a panel of dementia-relevant neuropsychological tests from an early-life baseline; were rated as looking older based on facial photographs; and reported themselves to be in worse health (*Belsky et al., 2015*). Subsequently, we reported that faster Pace of Aging is associated with early-life factors important for aging: familial longevity, low childhood social class, and adverse childhood experiences (*Belsky et al., 2017*), and that faster Pace of Aging is associated with older scores on Brain Age, a machine-learning-derived measure of structural MRI differences characteristic of different age groups (*Elliott et al., 2019*). Notably, Pace of Aging was not well-correlated with published epigenetic age clocks, which were designed to measure how old a person is rather than how fast they are aging biologically (*Belsky et al., 2018b*).

In Step 3, which we report here, we distill the Pace of Aging into a measurement that can be obtained from a single blood sample. Here we focused on blood DNA methylation as an accessible molecular measurement that is sensitive to changes in physiology occurring in multiple organ systems (*Birney et al., 2016*; *Bolund et al., 2017*; *Chambers et al., 2015*; *Chu et al., 2017*; *Hedman et al., 2017*; *Ma et al., 2019*; *Mill and Heijmans, 2013*; *Morris et al., 2017*; *Wahl et al., 2017*). We used data about the Pace of Aging from age 26 to 38 years in the Dunedin Study along with whole-genome methylation data at age 38 years. Elastic-net regression was applied to derive

an algorithm that captured DNA methylation patterns linked with variation among individuals in their Pace of Aging. The algorithm is hereafter termed 'DunedinPoAm'.

DunedinPoAm is qualitatively different from previously published DNA methylation measures of aging that were developed by comparing older individuals to younger ones. Those measures, often referred to as 'clocks,' are state measures. They estimate how much aging has occurred in an individual up to the point of measurement. DunedinPoAm is a rate measure. It is based on comparison of longitudinal change over time in 18 biomarkers of organ-system integrity among individuals who are all the same chronological age. DunedinPoAm estimates how fast aging is occurring during the years leading up to the time of measurement. Rather than a clock that records how much time has passed, DunedinPoAm is designed to function as a speedometer, recording how fast the subject is aging.

In Step 4, which we report here, we validated the DunedinPoAm in five ways. First, using the Dunedin Study, we tested if Study member's DunedinPoAm measured when they were aged 38 years could predict deficits in physical and cognitive functioning seven years later, when the cohort was aged 45 years. Second, we applied the DunedinPoAm algorithm to DNA methylation data from a second, cross-sectional, study of adults to evaluate patterning of DunedinPoAm by chronological age and sex and to test correlations of DunedinPoAm with self-reported health and proposed measures of biological age, including three epigenetic clocks. Third, we applied the DunedinPoAm algorithm to DNA methylation data from a third, longitudinal study of older men to test associations with chronic-disease morbidity and mortality. Fourth, we applied the DunedinPoAm algorithm to DNA methylation data from a fourth, longitudinal, study of young people to test if DunedinPoAm was accelerated by exposure to poverty and victimization, factors which are known to shorten healthy lifespan. Finally, to ascertain the potential usefulness of DunedinPoAm as a measure for trials of geroprotector treatments, we applied the algorithm to DNA methylation data from a randomized trial of caloric restriction, CALERIE (*Ravussin et al., 2015*). Earlier we reported from this trial that the intervention (two years of prescribed 25% caloric restriction) slowed the rate of biological aging as measured by a blood-chemistry biological-age composite measure (*Belsky et al., 2018a*). Here, using newly generated methylation data from blood drawn at the CALERIE baseline assessment, we tested if (a) DunedinPoAm from blood drawn before caloric restriction could predict the future rate of biological aging of participants during the two-year trial, and (b) if this prediction was disrupted in participants who underwent caloric restriction, but not among control participants.

We report promising results from this four-step research program, while appreciating that additional measurement development will be needed to support applied use of DunedinPoAm. A graphical illustration of our study design is presented in *Figure 1*.

## Results

### Capturing Pace of Aging in a single blood test

We derived the DunedinPoAm algorithm using data from Dunedin Study members for whom age-38 DNA methylation data were available (N = 810). We applied elastic-net regression (*Zou and Hastie, 2005*) using Pace of Aging between ages 26 to 38 years as the criterion. We included all methylation probes that appear on both the Illumina 450 k and EPIC arrays as potential predictor variables. We selected this overlapping set of probes for our analysis to facilitate application of the derived algorithm by other research teams using either array. We fixed the alpha parameter to 0.5, following the approach reported by *Horvath (2013)*. This analysis selected a set of 46 CpG sites (*Supplementary file 1A*). The 46-CpG elastic-net-derived DunedinPoAm algorithm, applied in the age-38 Dunedin DNA methylation data, was associated with the longitudinal 26–38 Pace of Aging measure (Pearson r = 0.56, *Figure 1—figure supplement 1*). This is likely an overestimate of the true out-of-sample correlation because the analysis is based on the same data used to develop the DunedinPoAm algorithm; bootstrap-cross-validation analysis estimated the out-of-sample correlation to be r = 0.33 (*Figure 1—figure supplement 2*).

### DunedinPoAm in midlife predicted future functional limitations
#### Physical functioning
As a primary criterion validity analysis of DunedinPoAm, we tested prospective associations of Dunedin Study members' age-38 DunedinPoAm values with their performance seven years later, when

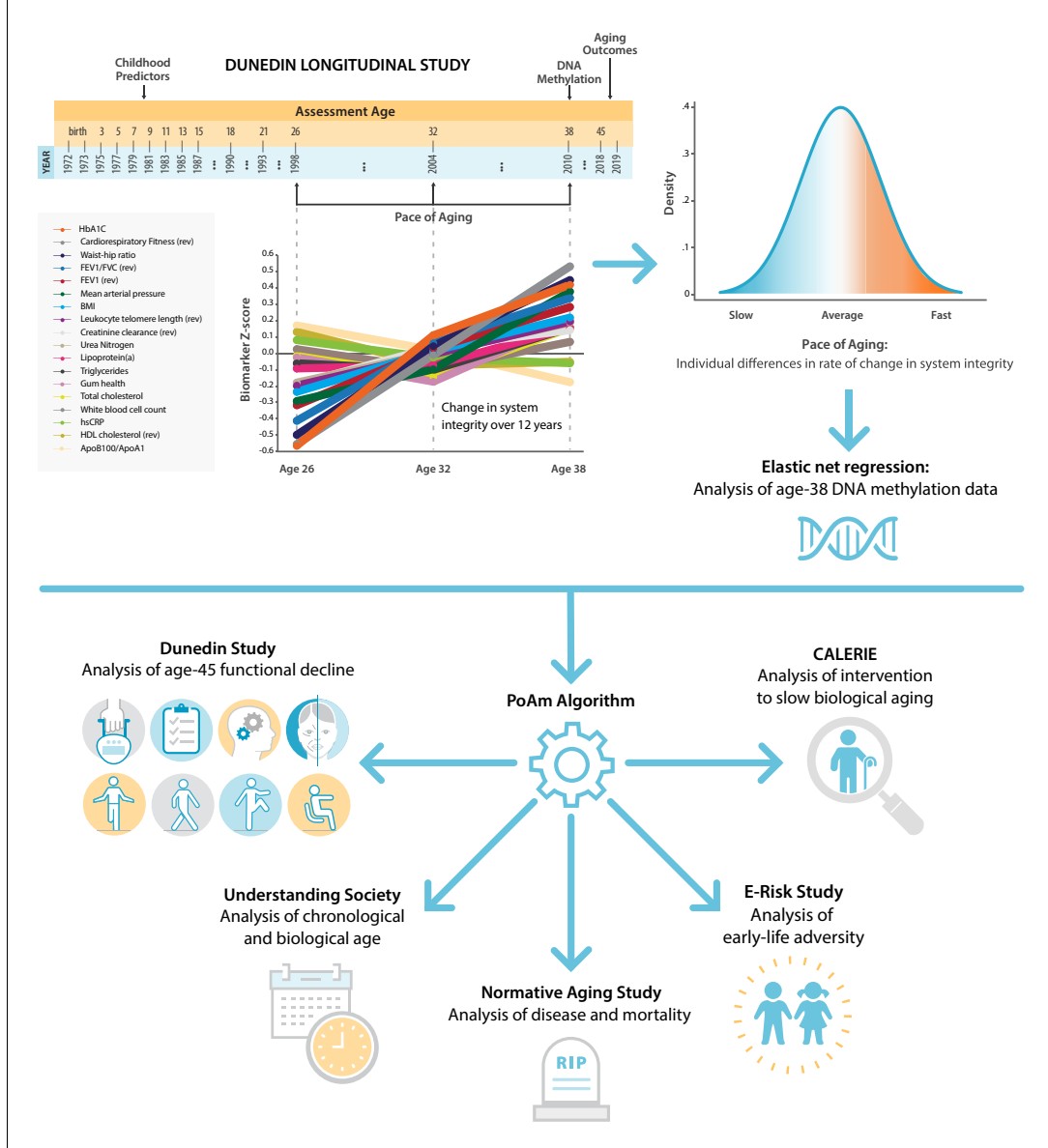

**Figure 1.** Schematic of design and follow-up of DunedinPoAm. DunedinPoAm is designed to quantify the rate of decline in system integrity experienced by an individual over the recent past; it functions like speedometer for the rate of aging. We developed DunedinPoAm from analysis of longitudinal change in 18 biomarkers of organ system integrity in the Dunedin Longitudinal Study birth cohort. Biomarkers were measured in 1998, 2004, and 2010, when all cohort members were aged 26, 32, and 38 years. We composited rates of change across the 18 biomarkers to produce a single measure of aging-related decline in system integrity, termed Pace of Aging. We then used elastic-net regression to develop a DNA-methylation predictor of Pace of Aging, called DunedinPoAm for Dunedin (P)ace (o)f (A)ging (m)ethylation. DNA methylation data for this analysis came from the age-38 assessment in 2010. We further evaluated the performance of DunedinPoAm using data from (a) the age-45 assessments of Dunedin Study members in 2018, (b) the Understanding Society Study, (c) the Normative Aging Study, (d) the E-risk Study, and (e) the CALERIE trial.
The online version of this article includes the following figure supplement(s) for figure 1:

**Figure supplement 1.** Association of DunedinPoAm with 12 year longitudinal Pace of Aging in the Dunedin Study.

**Figure supplement 2.** Bootstrap repetition analysis to estimate out-of-sample correlation between methylation Pace of Aging (mPoA) measures and longitudinal Pace of Aging in the Dunedin Study.

**Figure supplement 3.** Correlations of individual biomarker slopes with DunedinPoAm plotted against their correlations with 12 year longitudinal Pace of Aging.

they were aged 45 years, on tests of balance, walking speed, chair stands, grip strength, motor coordination, and Study-member reports about physical limitations. Performance scores were reversed so that positive correlations indicated an association between faster DunedinPoAm (i.e. higher values) and worse physical performance. Study members with faster DunedinPoAm at age 38 performed more poorly at age 45 on all physical performance tests, with the exception of grip strength, and reported more functional limitations (standardized effect-sizes for tests of balance, walking speed, chair stands, and physical limitations r = 0.15–0.29, p<0.001 for all; grip strength r = 0.05, 95% CI [−0.02–0.12], p=0.162). Effect-sizes are graphed in the dark blue bars in *Figure 2*, Panel A.

## Physical decline

For balance, grip strength, motor coordination, and functional limitations, the Dunedin study administered the same assessments at the age-38 and age-45 assessments. We used these data to measure change in physical function across the 7 year interval. We computed change scores by subtracting the age-45 score from the age-38 score. Change scores for balance and physical limitations indicated worsening of physical functioning across the 7 year interval (in terms of age-38 standard-deviation units (SDs): balance declined by 0.26 SDs 95% CI [0.19–0.33], motor coordination declined by 0.16 [0.10–0.22] SDs, and physical limitations increased by 0.26 [0.18–0.33] SDs). In contrast, grip strength increased slightly (0.05 [0.00–0.11] SDs). Study members with faster age-38 DunedinPoAm experienced greater decline in balance at age 45 (r = 0.11 [0.04–0.19], p=0.008) and a greater increase in physical limitations (r = 0.10 [0.02–0.18], p=0.012). The association with change in motor coordination was consistent in direction, but was not statistically different from zero at the alpha = 0.05 threshold (r = 0.06 [−0.14–0.01], p=0.109). There was no association between DunedinPoAm and change in grip strength (r = 0.00 [−0.07–0.08], p=0.897). Effect-sizes are graphed in the top two rows of *Figure 2*, Panel B.

## Cognitive functioning

We evaluated cognitive functioning from tests of perceptual reasoning, working memory, and processing speed, which are known to show aging related declines already by the fifth decade of life (*Hartshorne and Germine, 2015*; *Park et al., 2002*). Study members with faster age-38 DunedinPoAm performed more poorly on all age-45 cognitive tests (r = 0.14–0.28, p<0.001 for all). Effect-sizes are graphed in the light-blue bars of *Figure 2*, Panel A.

## Cognitive decline

Cognitive functioning in early life is a potent risk factor for chronic disease and dementia in later life and for accelerated aging in midlife (*Belsky et al., 2017*; *Deary and Batty, 2006*). Therefore, to evaluate whether associations between DunedinPoAm and cognitive test performance at age 45 might reflect reverse causation instead of early cognitive decline, we next conducted analysis of cognitive decline between adolescence and midlife. We evaluated cognitive decline by comparing Study-members' cognitive-test performance at age 45 to their cognitive-test performance three decades earlier when they were ages 7–13 years. Cognitive performance was measured from composite scores on the Wechsler Intelligence Scales (the Wechsler Adult Intelligence Scales Version IV at the age-45 assessment and the Wechsler Intelligence Scales for Children Version R at the earlier time-points). On average, Study members showed a decline of 2.00 IQ points (95% CI [1.31–2.70]) across the follow-up interval. We conducted two analyses to test if participants with faster DunedinPoAm experienced more cognitive decline. First, we computed difference scores (age-45 IQ – childhood baseline IQ) and regressed these difference scores on DunedinPoAm. Second, we conducted analysis of residualized change by regressing age-45 IQ on DunedinPoAm and childhood IQ. Both analyses found that Study members with faster DunedinPoAm experienced more decline (difference-score r = 0.12, [0.05–0.19], p=0.001; residualized change r = 0.20 [0.13–0.27]). The DunedinPoAm association with cognitive decline is graphed in the third row of *Figure 2*, Panel B.

## Subjective signs of aging

We evaluated subjective signs of aging from Study members' ratings of their current health status (excellent, very-good, good, fair, poor) and from ratings of perceived age made by undergraduate raters based on facial photographs. Study members with faster age-38 DunedinPoAm rated

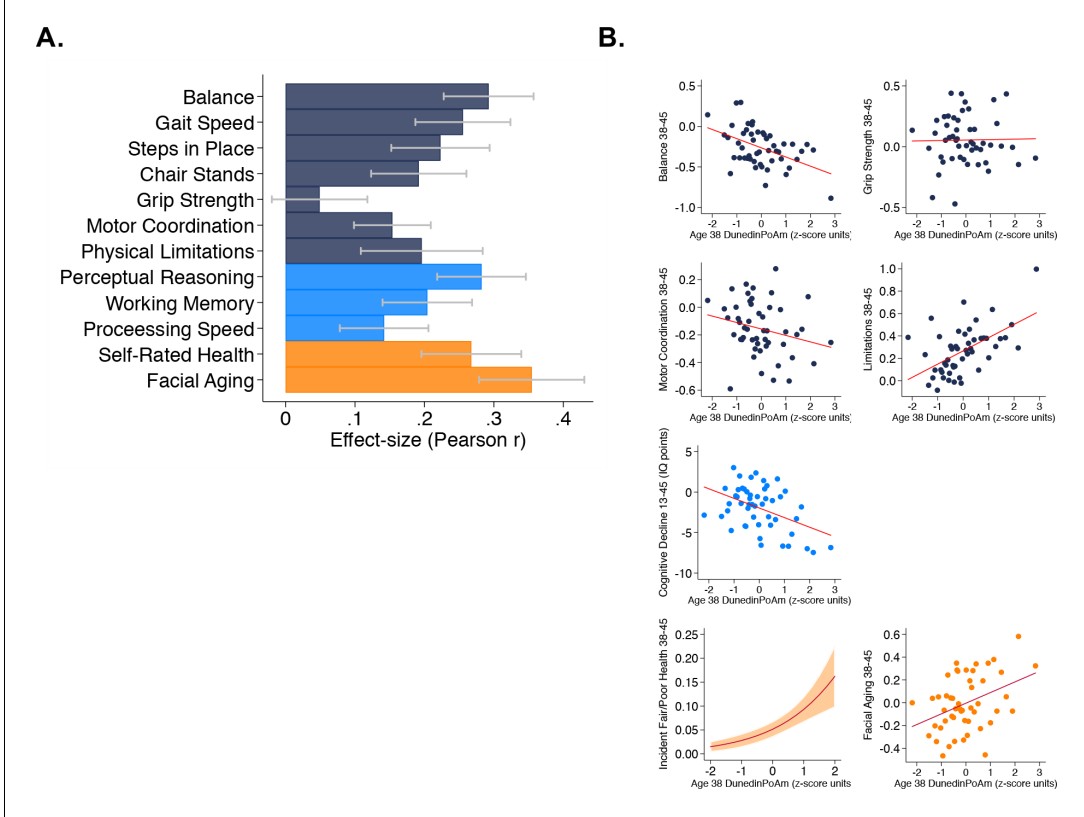

**Figure 2.** Faster age-38 DunedinPoAm is associated with poorer physical and cognitive functioning and subjective signs of aging at age 45 years, and with physical, cognitive, and subjective decline in the Dunedin Study. Panel A graphs effect-sizes for age-38 DunedinPoAm associations with age-45 measures of physical and cognitive functioning and subjective signs of aging in the Dunedin Study. Effect-sizes are standardized regression coefficients interpretable as Pearson r. Models included covariate adjustment for sex. Panel B graphs associations between DunedinPoAm and change in physical functioning between age 38 and age 45 (top two rows), change in cognitive functioning between age 13 and age 45 (third row), and incident fair/poor health and accelerated facial aging between ages 38 and 45 (bottom row). Graphs for changes in balance, grip-strength, physical limitations, cognition, and facial aging are binned scatterplots. Plotted points reflect average x- and y-coordinates for 'bins' of approximately ten Study members. Fitted slopes show the association estimated from the raw, un-binned data. The y-axis scale on graphs of balance, grip-strength, and physical limitations shows change scores (age 45 – age 38) scaled in terms of age-38 standard deviation units. The y-axis scale on the graph of cognitive change shows the difference in IQ score (age 45 – baseline). The graph of change in facial aging shows the change in z-score between measurement intervals (age 45 – age 38). Models included covariate adjustment for sex. The graph for self-rated health plots the predicted probability (fitted slope) and 95% confidence interval (shaded area) of incident fair/poor health at age 45. The effect-size reported on the graph is the incidence-rate ratio (IRR) associated with a 1-SD increase in DunedinPoAm estimated from Poisson regression. The model included covariate adjustment for sex.

The online version of this article includes the following figure supplement(s) for figure 2:

**Figure supplement 1.** Association of DunedinPoAm and 12 year longitudinal Pace of Aging with age-45 healthspan characteristics in the Dunedin Study.

**Figure supplement 2.** Effect-sizes for association of DunedinPoAm and epigenetic clocks with age-45 healthspan characteristics in the Dunedin Study.

**Figure supplement 3.** Effect-sizes for association of DunedinPoAm and epigenetic clocks with change over time in healthspan characteristics in the Dunedin Study.

themselves to be in worse health at age 45 (r = 0.27, 95% CI [0.20–0.34]). These Study members were also rated as looking older (r = 0.35 [0.28–0.43]). Effect-sizes are graphed in the orange bars in *Figure 2*, Panel A.

## Subjective signs of decline with aging

We next analyzed change in subjective signs of aging. Across the 7 year follow-up interval, an increasing number of Study members rated themselves as being in fair or poor health (6% rated their health as fair or poor at age 38, as compared to 8% 7 years later at age 45). Those with faster age-38 DunedinPoAm were more likely to transition to the fair/poor categories (Incidence Rate Ratio

(IRR) = 1.79 95% CI [1.48–2.18]). We tested if Study members with faster DunedinPoAm experienced more rapid facial aging by subtracting the age-45 score from the age-38 score and regressing this difference on DunedinPoAm. This analysis tested if Study members with faster DunedinPoAm experienced upward rank mobility within the cohort in terms of how old they looked. Study members with faster age-38 DunedinPoAm were rated as looking older relative to peers at age 45 than they had been at age 38 (r = 0.10 [0.03–0.18]). Effect-sizes are graphed in the bottom row of *Figure 2*, Panel B.

## Comparing DunedinPoAm versus Pace of Aging

We compared DunedinPoAm effect-sizes to effect-sizes for the original, 18-biomarker 3-time-point Pace of Aging. Across the domains of physical function, cognitive function, and subjective signs of aging, DunedinPoAm effect-sizes were similar to and sometimes larger than effect-sizes for the original Pace of Aging measure (*Supplementary file 1B*, *Figure 2—figure supplement 1*).

Covariate adjustment to models for estimated cell counts (*Houseman et al., 2012*) did not change results. Covariate adjustment for smoking history at age 38 years modestly attenuated some effect-sizes and attenuated DunedinPoAm associations with cognitive decline to near zero. Results for all models are reported in *Supplementary file 1C*.

In comparison to DunedinPoAm, effect-sizes for associations with functional limitations were smaller for the Horvath, Hannum, and Levine epigenetic clocks and, in the cases of the Horvath and Hannum clocks, were not statistically different from zero at the alpha = 0.05 threshold for most outcomes. Effect-sizes are reported in *Supplementary file 1C* and plotted in *Figure 2—figure supplements 2* and *3*.

## Evaluating DunedinPoAm and other methylation clocks in the Understanding Society Study

To test variation in DunedinPoAm and to compare it with published methylation measures of biological aging, we conducted analysis using data on N = 1175 participants aged 28–95 years (M = 58, SD = 15; 42% male) in the UK Understanding Society Study. In this mixed-age sample, the mean DunedinPoAm was 1.03 years of biological aging per each calendar year (SD = 0.07).

We first tested if higher DunedinPoAm levels, which indicate faster aging, were correlated with older chronological age. Mortality rates increase with advancing chronological age, although there may be some slowing at the oldest ages (*Barbi et al., 2018*). This suggests the hypothesis that the rate of aging increases across much of the adult lifespan. Consistent with this hypothesis, Understanding Society participants who were of older chronological age tended to have faster DunedinPoAm (r = 0.11, [0.06–0.17], p<0.001; *Figure 3* Panel A). We also compared DunedinPoAm with three methylation measures of biological age: the epigenetic clocks proposed by Horvath, Hannum, and Levine (*Hannum et al., 2013*; *Horvath, 2013*; *Levine et al., 2018*). These epigenetic clocks were highly correlated with chronological age in the Understanding Society sample (Horvath Clock r = 0.91, Hannum Clock r = 0.92, Levine Clock r = 0.88).

Next, to test if DunedinPoAm captured similar information about aging to published epigenetic clocks, we regressed each of the published clocks on chronological age and predicted residual values, following the procedure used by the developers of the clocks. These residuals are referred to in the literature as measures of 'epigenetic age acceleration.' None of the 46 CpGs included in the DunedinPoAm algorithm overlapped with CpGs in these epigenetic clocks. Nevertheless, DunedinPoAm was moderately correlated with epigenetic age acceleration measured from the clocks proposed by Hannum (r = 0.24) and Levine (r = 0.30). DunedinPoAm was less-well correlated with acceleration measured from the Horvath clock (r = 0.06). Associations among DunedinPoAm and the epigenetic clocks in the Understanding Society sample are shown in *Figure 3* Panel B.

Finally, we tested correlations of DunedinPoAm with (a) a measure of biological age derived from blood chemistry and blood pressure data, and (b) a measure of self-rated health. We computed biological age from Understanding Society blood chemistry and blood pressure data following the Klemera and Doubal method (KDM) (*Klemera and Doubal, 2006*) and the procedure described by *Levine (2013)*. KDM Biological Age details are reported in the Materials and Methods. Participants with faster DunedinPoAm had more advanced KDM Biological Age (r = 0.20 95% CI [0.15–0.26], p<0.001;) and worse self-rated health (r = −0.22 [-0.28,–0.16], p<0.001;). Covariate adjustment to

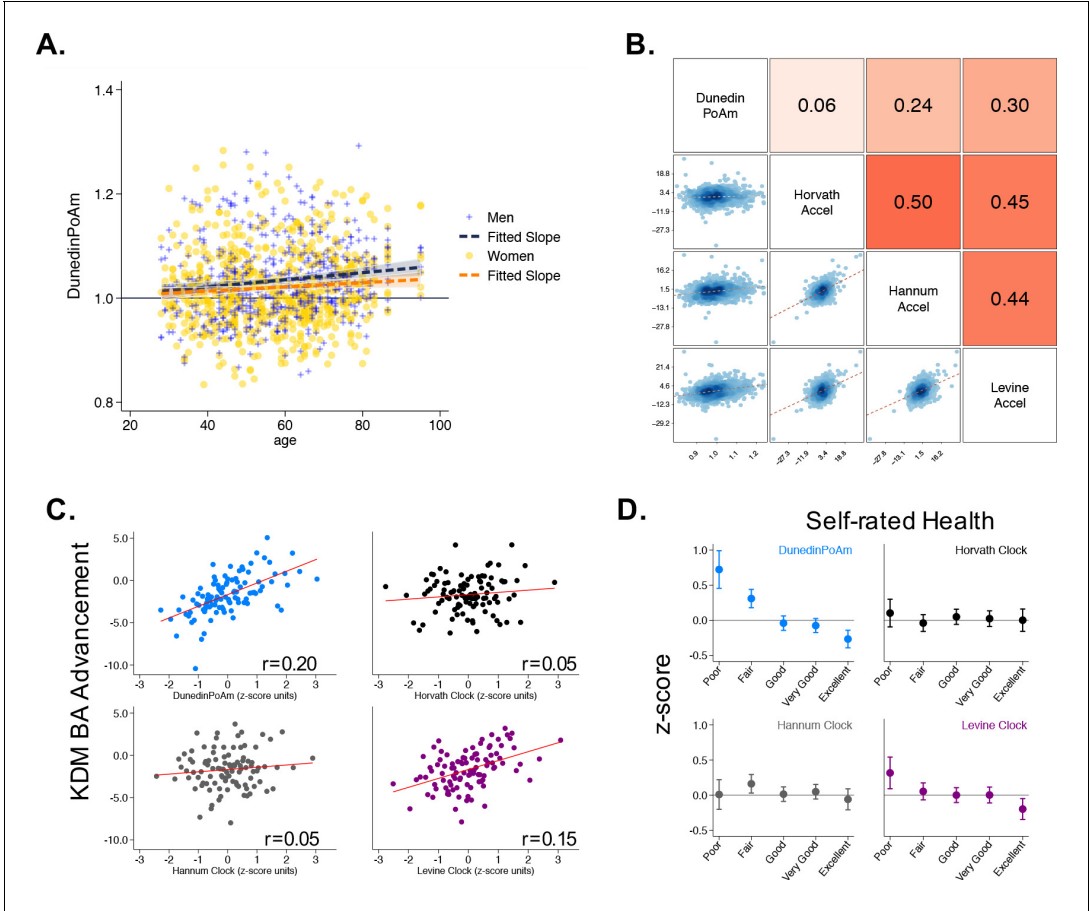

**Figure 3.** Associations of DunedinPoAm with chronological age, epigenetic clocks, KDM Biological Age, and self-rated health in Understanding Society. Panel A shows a scatterplot and fitted slopes illustrating the association between chronological age (x-axis) and DunedinPoAm (y-axis) in women and men in the Understanding Society sample. Data for women are plotted with yellow dots (orange slope) and for men with blue crosses (navy slope). The figure illustrates a positive association between chronological age and DunedinPoAm (Pearson r = 0.11 95% CI [0.06–0.17]). Panel B shows a matrix of correlations and association plots among DunedinPoAm and age-acceleration residuals of the Horvath, Hannum, and Levine epigenetic clocks. The diagonal cells of the matrix list the DNA methylation measures. The half of the matrix below the diagonal shows scatter plots of associations. For each scatter-plot cell, the y-axis corresponds to the variable named along the matrix diagonal to the right of the plot and the x-axis corresponds to the variable named along the matrix diagonal above the plot. The half of the matrix above the diagonal lists Pearson correlations between the DNA methylation measures. For each correlation cell, the value reflects the correlation of the variables named along the matrix diagonal to the left of the cell and below the cell. Panel C graphs binned scatterplots of associations of DunedinPoAm and epigenetic clocks with KDM Biological Age advancement (the difference between KDM Biological Age and chronological age). Each plotted point shows average x- and y-coordinates for 'bins' of approximately 50 participants. Regression slopes are graphed from the raw, un-binned data. Panel D plots average values of the DNA methylation variables by Understanding Society participants' self-rated health status. Error bars show 95% confidence intervals.
The online version of this article includes the following figure supplement(s) for figure 3:

**Figure supplement 1.** Effect-sizes for associations of DunedinPoAm and epigenetic clocks with KDM Biological Age and self-rated health in Understanding Society.

models for estimated cell counts (*Houseman et al., 2012*) and smoking status did not change results. Results for all models are reported in *Supplementary file 1D*.

In comparison to DunedinPoAm, effect-sizes for associations with self-rated health and KDM Biological Age were smaller for the epigenetic clocks and, in the cases of the Horvath and Hannum clocks, were not statistically different from zero at the alpha = 0.05 threshold (*Figure 3* Panels C and D). Effect-sizes are reported in *Supplementary file 1D* and plotted in *Figure 3—figure supplement 1*.

## DunedinPoAm was associated with chronic disease morbidity and increased risk of mortality among older men in the Normative Aging Study (NAS)

To test if faster DunedinPoAm was associated with morbidity and mortality, we analyzed data from N = 771 older men in the Veterans Health Administration Normative Aging Study (NAS; at baseline, mean chronological age = 77, SD = 7).

We first tested if higher DunedinPoAm levels, which indicate faster aging, were associated with increased risk of mortality. During follow-up from 1999 to 2013, 46% of NAS participants died over a mean follow-up of 7 years (SD = 7). Those with faster DunedinPoAm at baseline were at increased risk of death (Hazard Ratio (HR) = 1.29 [1.16–1.45], p<0.001; *Figure 4*).

We next tested if NAS participants with faster DunedinPoAm experienced higher levels of chronic disease morbidity, measured as the count of diagnosed diseases (hypertension, type-2 diabetes, cardiovascular disease, chronic obstructive pulmonary disease, chronic kidney disease, and cancer). During follow-up across 4 assessments during 1999–2013 (n = 1448 observations of the N = 771 participants), n = 175 NAS participants were diagnosed with a new chronic disease. Those with faster baseline DunedinPoAm were at increased risk of new diagnosis (HR = 1.19 [1.03–1.38], p<0.019). In repeated-measures analysis of prevalent chronic disease, faster DunedinPoAm was associated with having a higher level of chronic disease morbidity (IRR = 1.15 [1.11–1.20], p<0.001).

Finally, we utilized the repeated-measures data to test if NAS participants' DunedinPoAms increased as they aged. We tested within-person change in DunedinPoAm over time (n = 1253 observations of N = 536 participants with 2–4 timepoints of DNA methylation data). Consistent with Understanding Society analysis showing faster DunedinPoAm in older as compared to younger adults, NAS participants' DunedinPoAm values increased across repeated assessments. For every five years of follow-up, participants' DunedinPoAms increased by 0.012 (SE = 0.002, p<0.001) units, or about 0.2 standard deviations.

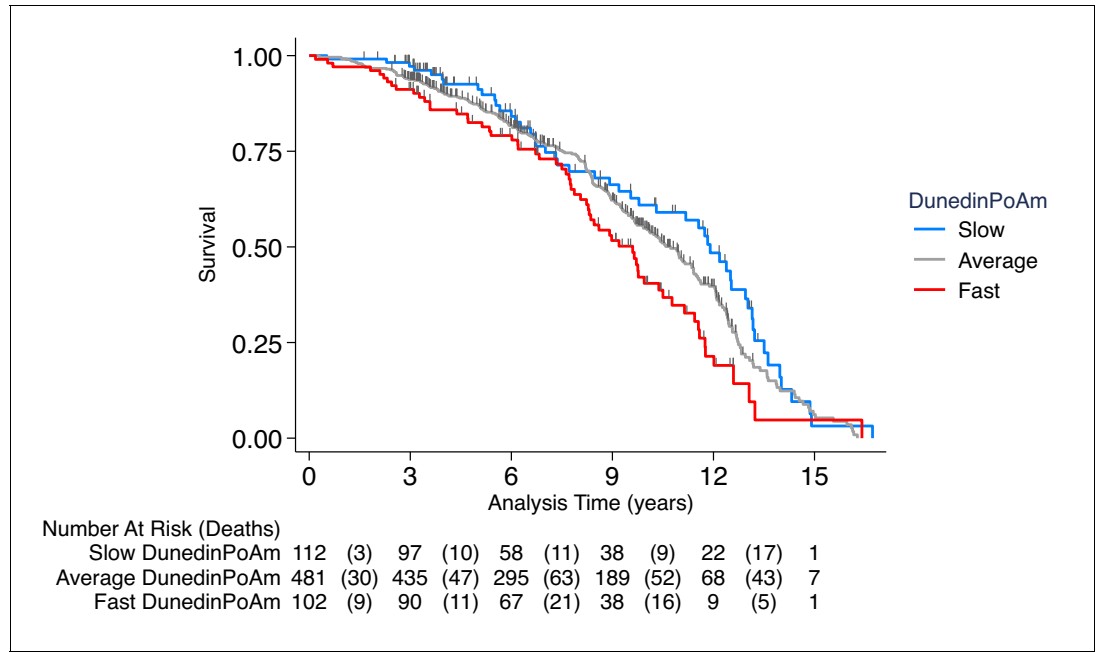

| Number At Risk (Deaths) | | | | | | | | | | |
|---|---|---|---|---|---|---|---|---|---|---|
| Slow DunedinPoAm | 112 | (3) | 97 | (10) | 58 | (11) | 38 | (9) | 22 | (17) | 1 |
| Average DunedinPoAm | 481 | (30) | 435 | (47) | 295 | (63) | 189 | (52) | 68 | (43) | 7 |
| Fast DunedinPoAm | 102 | (9) | 90 | (11) | 67 | (21) | 38 | (16) | 9 | (5) | 1 |

**Figure 4.** Association of DunedinPoAm with mortality in the Normative Aging Study (NAS). The figure plots Kaplan-Meier curves for three groups of NAS participants: those with DunedinPoAm 1 SD or more below the mean ('slow' DunedinPoAm, blue line); those with DunedinPoAm within 1 SD of the mean ('average' DunedinPoAm, gray line); and those with DunedinPoAm 1 SD or more above the mean ('fast' DunedinPoAm, red line). Censoring of participants prior to death is indicated with a vertical gray hash mark. The table below the figure details the number of participants at risk per 3 year interval and, in parentheses, the number who died during the interval. The number censored can be calculated by subtracting the number of deaths in an interval from the difference between the number at risk in that interval and the number at risk in the following interval.

The online version of this article includes the following figure supplement(s) for figure 4:

**Figure supplement 1.** Effect-sizes for association of DunedinPoAm and epigenetic clocks with mortality and morbidity in the Normative Aging Study.

Covariate adjustment to models for estimated cell counts (*Houseman et al., 2012*) and smoking status did not change results, with the exception that the effect-size for DunedinPoAm was attenuated below the alpha = 0.05 threshold of statistical significance in smoking-adjusted analysis of chronic disease incidence. Results for all models are reported in *Supplementary file 1E*.

In comparison to DunedinPoAm, effect-sizes for associations with mortality and chronic disease were smaller for the epigenetic clocks and were not statistically different from zero in many of the models (*Supplementary file 1E* and *Figure 4—figure supplement 1*).

## Childhood exposure to poverty and victimization were associated with faster DunedinPoAm in young adults in the E-Risk Study

To test if DunedinPoAm indicated faster aging in young people with histories of exposure thought to shorten healthy lifespan, we analyzed data from N = 1658 members of the E-Risk Longitudinal Study. The E-Risk Study follows a 1994–95 birth cohort of same-sex twins. Blood DNA methylation data were collected when participants were aged 18 years. We analyzed two exposures associated with shorter healthy lifespan, childhood low socioeconomic status and childhood victimization. Socioeconomic status was measured from data on parents' education, occupation, and income (*Trzesniewski et al., 2006*). Victimization was measured from exposure dossiers compiled from interviews with the children's mothers and home-visit assessments conducted when the children were aged 5, 7, 10, and 12 (*Fisher et al., 2015*). The dossiers recorded children's exposure to domestic violence, peer bullying, physical and sexual harm by an adult, and neglect. 72% of the analysis sample had no victimization exposure, 21% had one type of victimization exposure, 4% had two types of exposure, and 2% had three or more types of exposure.

E-Risk adolescents who grew up in lower socioeconomic-status families exhibited faster Dunedin-PoAm (Cohen's d for comparison of low to moderate SES = 0.21 [0.06–0.35]; Cohen's d for comparison of low to high SES = 0.44 [0.31–0.56]; Pearson r = 0.19 [0.13–0.24]). In parallel, E-Risk adolescents with exposure to more types of victimization exhibited faster DunedinPoAm (Cohen's d for comparison of never victimized to one type of victimization = 0.28 [0.15–0.41]; Cohen's d for comparison of never victimized to two types of victimization = 0.48 [0.23–0.72]; Cohen's d for comparison of never victimized to three or more types of victimization = 0.53 [0.25–0.81]; Pearson r = 0.15 [0.10–0.20]). Covariate adjustment to models for estimated cell counts (*Houseman et al., 2012*) did not change results. Adjustment for smoking status attenuated effect-sizes by about half, but most associations remained statistically different from zero at the alpha = 0.05 level. Results for all models are reported in *Supplementary file 1F*. Differences in DunedinPoAm across strata of childhood socioeconomic status and victimization are graphed in *Figure 5*.

In comparison to DunedinPoAm, effect-sizes for associations with childhood socioeconomic circumstances and victimization were smaller for the epigenetic clocks and, in the cases of the Horvath and Hannum clocks, were not statistically different from zero at the alpha = 0.05 threshold. Effect-sizes are reported in *Supplementary file 1F* and plotted in *Figure 5—figure supplement 1*.

## DunedinPoAm measured at baseline in the CALERIE randomized trial predicted future rate of aging measured from clinical-biomarker data

The CALERIE Trial is the first randomized trial of long-term caloric restriction in non-obese adult humans. CALERIE randomized N = 220 adults on a 2:1 ratio to treatment of 25% caloric restriction (CR-treatment) or control ad-libitum (AL-control, as usual) diet for two years (*Ravussin et al., 2015*). We previously reported that CALERIE participants who were randomized to CR-treatment experienced a slower rate of biological aging as compared to participants in the AL-control arm based on longitudinal change analysis of clinical-biomarker data from the baseline, 12 month, and 24 month follow-up assessments (*Belsky et al., 2018a*). Among control participants, the rate of increase in biological age measured using the Klemera-Doubal method (KDM) Biological Age algorithm was 0.71 years of biological age per 12 month follow-up interval. (This slower-than-expected rate of aging could reflect differences between CALERIE Trial participants, who were selected for being in good health, and the nationally representative NHANES sample in which the KDM algorithm was developed [*Belsky et al., 2018a*].) In contrast, among treatment participants, the rate of increase was only 0.11 years of biological age per 12 month follow-up interval (difference b = −0.60 [-0.99,–0.21]). We subsequently generated DNA methylation data from blood DNA that was collected at

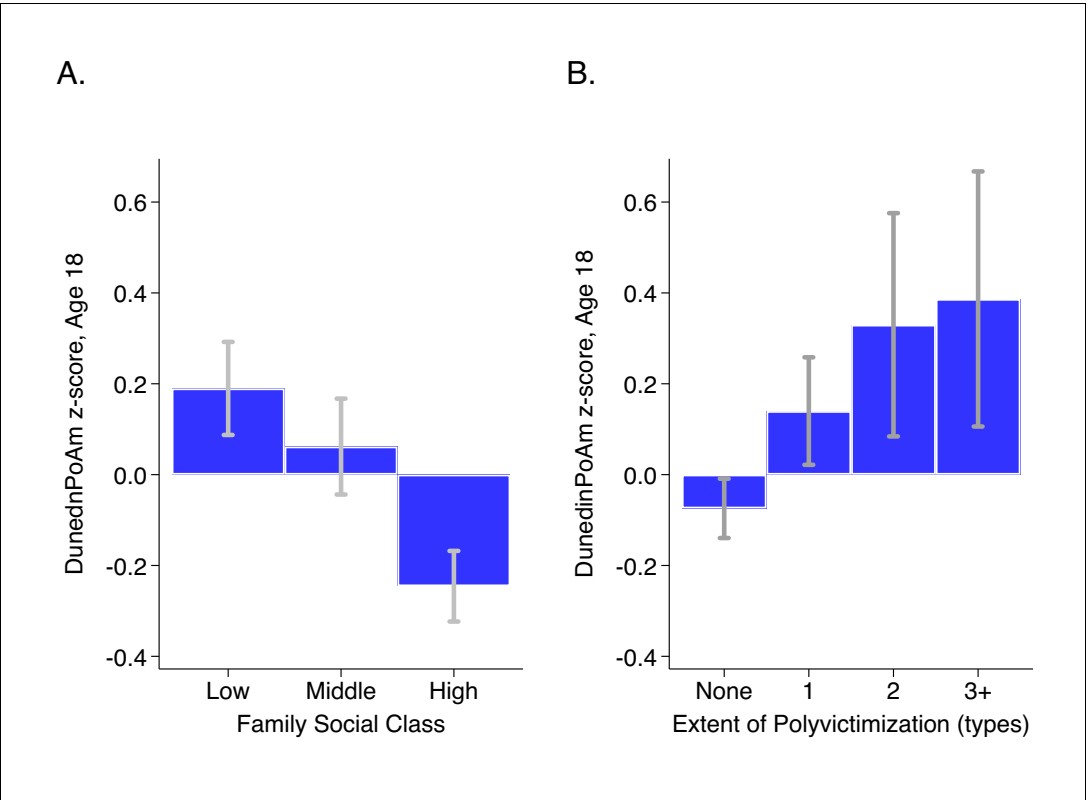

**Figure 5.** DunedinPoAm levels by strata of childhood socioeconomic status (SES) and victimization in the E-Risk Study. Panel A (left side) plots means and 95% CIs for DunedinPoAm measured at age 18 among E-Risk participants who grew up low, middle, and high socioeconomic status households. Panel B (right side) plots means and 95% CIs for DunedinPoAm measured at age 18 among E-Risk participants who experienced 0, 1, 2, or three or more types of victimization through age 12 years.

The online version of this article includes the following figure supplement(s) for figure 5:

**Figure supplement 1.** Effect-sizes for associations of childhood socioeconomic status and victimization exposure with DunedinPoAm and epigenetic clocks.

the baseline assessment of the CALERIE trial for a sub-sample (N = 68 AL-control participants and 118 CR-treatment participants). We used these methylation data to calculate participants' Dunedin-PoAm values at study baseline. We then tested if baseline DunedinPoAm could predict participants' future rate of biological aging as they progressed through the trial.

We first replicated our original analysis within the methylation sub-sample. Results were the same as in the full sample (*Supplementary file 1G*). Next, we compared DunedinPoAm between CR-treatment and AL-control participants. As expected, there was no group difference at baseline (AL M = 1.00, SD = 0.05; CR M = 1.01, SD = 0.06, p-value for difference = 0.440). Finally, we tested if participants' baseline DunedinPoAm was associated with their rate of biological aging over the 24 months of follow-up, and if this association was modified by randomization to caloric restriction as compared to ad libitum diet. For AL-control participants, faster baseline DunedinPoAm predicted faster biological aging over the 24 month follow-up, although in this small group this association was not statistically significant at the alpha = 0.05 level (b = 0.22 [-0.05, 0.49], p=0.104). For CR-treatment participants, the association of baseline DunedinPoAm with future rate of aging was sharply reduced, (b = −0.08 [-0.24, 0.09], p=0.351), although the difference between the rate of aging in the AL-control and CR-treatment groups did not reach the alpha = 0.05 threshold for statistical significance (interaction-term testing difference in slopes b = −0.30 [-0.61, 0.01], p-value=0.060). Slopes of change in KDM Biological Age for participants in the AL-control and CR-treatment groups are plotted for fast baseline DunedinPoAm (1 SD above the mean) and slow baseline DunedinPoAm

(1 SD below the mean) in *Figure 6*. CALERIE DNA methylation data are not yet available to test if the intervention altered post-treatment DunedinPoAm.

## Discussion

Breakthrough discoveries in the new field of geroscience suggest opportunities to extend healthy life-span through interventions that slow biological processes of aging (*Campisi et al., 2019*). To advance translation of these interventions, measures are needed that can detect changes in a person's rate of biological aging (*Moffitt et al., 2017*). We previously showed that the rate of biological aging can be measured by tracking change over time in multiple indicators of organ-system integrity (*Belsky et al., 2015*). Here, we report data illustrating the potential to streamline measurement of Pace of Aging to an exportable, inexpensive and non-invasive blood test, and thereby ease implementation of Pace of Aging measurement in studies of interventions to slow processes of biological aging.

We conducted machine-learning analysis of the original Pace of Aging measure using elastic-net regression and whole-genome blood DNA methylation data. We trained the algorithm to predict how fast a person was aging. We called the resulting algorithm 'DunedinPoAm' for ' (P)ace (o)f (A) ging (m)ethylation'. There were four overall findings:

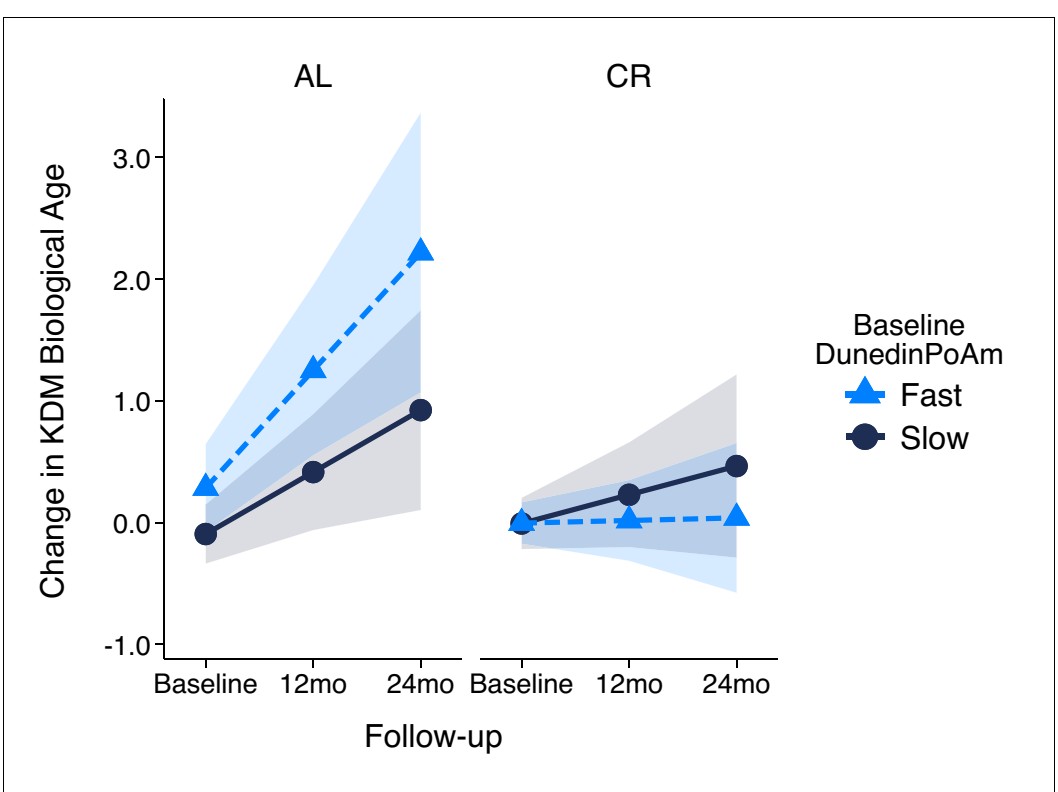

**Figure 6.** Change in KDM Biological Age over 24 month follow-up by treatment condition and baseline DunedinPoAm in the CALERIE Trial. Fast DunedinPoAm is defined as 1 SD above the sample mean. Slow DunedinPoAm is defined as 1 SD below the sample mean. Slopes are predicted values from mixed effects regression including a 3-way interaction between trial condition, time, and continuous DunedinPoAm at baseline. The figure shows that in the Ad Libitum (AL) arm of the trial, participants with fast DunedinPoAm at baseline experience substantially more change in KDM Biological Age from baseline to follow-up as compared to AL participants with slow DunedinPoAm. In contrast, there was little difference between participants with fast as compared to slow DunedinPoAm in the Caloric Restriction (CR) arm of the trial.

The online version of this article includes the following figure supplement(s) for figure 6:

**Figure supplement 1.** Effect-sizes for associations of DunedinPoAm and epigenetic clocks with future rate of change in KDM Biological Age in Ad Libitum (AL) control group and Caloric Restriction (CR) intervention group participants in the CALERIE Trial.

First, while DunedinPoAm was not a perfect proxy of Pace of Aging, it nevertheless captured critical information about Dunedin Study members' healthspan-related characteristics. Across the domains of physical function, cognitive function, and subjective signs of aging, Study members with faster DunedinPoAm at age 38 were worse off seven years later at age 45 and, in repeated-measures analysis of change, they showed signs of more rapid decline. Effect-sizes were equal to or greater than those for the 18-biomarker 3-time point measure of Pace of Aging. This result suggests that the DNA-methylation elastic-net regression used to develop DunedinPoAm may have distilled the aging signal from the original Pace of Aging measure and excluded some noise. In sum, DunedinPoAm showed promise as an easy-to-implement alternative to Pace of Aging. Emerging technologies for deep-learning analysis (*Zhavoronkov et al., 2019*) may improve methylation measurement of Pace of Aging. Alternatively, integration of methylation data with additional molecular datasets (*Hasin et al., 2017*; *Zierer et al., 2015*) may be needed to achieve precise measurement of Pace of Aging from a single time-point blood sample.

Second, DunedinPoAm analysis of the Understanding Society and NAS samples provided proof-of-concept for using DunedinPoAm to quantify biological aging. Age differences in DunedinPoAm parallel population demographic patterns of mortality risk. In the Understanding Society sample, older adults had faster DunedinPoAm as compared to younger ones. In the NAS sample, participants' DunedinPoAm values increased as they aged. These observations are consistent with the well-documented acceleration of mortality risk with advancing chronological age (*Robine, 2011*). However, it sets DunedinPoAm apart from other indices of biological aging, which are not known to register this acceleration (*Finch and Crimmins, 2016*; *Li et al., 2020*). DunedinPoAm may therefore provide a novel tool for testing how the rate of aging changes across the life course and whether, as demographic data documenting so-called 'mortality plateaus' suggest, processes of aging slow down at the oldest chronological ages (*Barbi et al., 2018*).

DunedinPoAm is related to but distinct from alternative approaches to quantification of biological aging. DunedinPoAm was moderately correlated with aging rates measured by the epigenetic clocks proposed by *Hannum et al. (2013)*; *Levine et al. (2018)* as well as KDM Biological Age derived from clinical biomarker data (*Klemera and Doubal, 2006*; *Levine, 2013*), and with self-rated health. Consistent with findings for the measured Pace of Aging (*Belsky et al., 2018b*), DunedinPoAm was only weakly correlated with the multi-tissue clock proposed by Horvath. DunedinPoAm was more strongly correlated with a clinical-biomarker measure of biological age, with self-rated health, with functional test-performance and decline, and with morbidity and mortality as compared to the epigenetic clocks.

Third, DunedinPoAm is already variable by young adulthood and is accelerated in young people at risk for eventual shortened healthspan. E-Risk young adults who grew up in socioeconomically disadvantaged families or who were exposed to victimization early in life already showed accelerated DunedinPoAm by age 18, consistent with epidemiological observations of shorter healthy lifespan for individuals with these exposures (*Adler and Rehkopf, 2008*; *Danese and McEwen, 2012*). We previously found that Dunedin Study members with histories of early-life adversity showed accelerated Pace of Aging in their 30 s (*Belsky et al., 2017*). DunedinPoAm analysis of the E-Risk cohort suggests effects may be already manifest at least a decade earlier. DunedinPoAm may therefore provide a useful index that can be applied to evaluate prevention programs to buffer at-risk youth against health damaging effects of challenging circumstances.

Fourth, DunedinPoAm analysis of the CALERIE trial provided proof-of-concept for using DunedinPoAm to quantify biological aging in geroprotector intervention studies. DunedinPoAm measures the rate of aging over the recent past. Control-arm participants' baseline DunedinPoAm correlated positively with their clinical-biomarker pace of aging over the two years of the trial, consistent with the hypothesis that their rate of aging was not altered. In contrast, there was no relationship between DunedinPoAm and clinical-biomarker pace of aging for caloric-restriction-arm participants, consistent with the hypothesis that caloric restriction altered participants' rate of aging. Ultimately, data on DunedinPoAm for all CALERIE participants (and participants in other geroprotector trails) at trial baseline and follow-up will be needed to establish utility of DunedinPoAm as a surrogate endpoint. In the mean-time, these data establish potential to use DunedinPoAm as a pre-treatment covariate in geroprotector trials to boost statistical power (*Kahan et al., 2014*) or to screen participants for enrollment, for example to identify those who are aging more rapidly and may therefore show larger effects of treatment.

We acknowledge limitations. Foremost, DunedinPoAm is a first step toward a single-assay cross-sectional measurement of Pace of Aging. The relatively modest size of the Dunedin cohort and the lack of other cohorts that have the requisite three or more waves of repeated biomarkers to measure the Pace of Aging limited sample size for our machine-learning analysis to develop methylation algorithms. As Pace of Aging is measured in additional cohorts, more refined analysis to develop DunedinPoAm-type algorithms will become possible. A related issue is scaling of DunedinPoAm. The original Pace of Aging measure from which DunedinPoAm was developed is denominated in 'years' of physiological decline occurring per 12 months of calendar time. Units of DunedinPoAm can, in principle, be interpreted in the same way. But replication in additional cohorts is needed. In addition, our work thus far has not addressed population diversity in biological aging. The Dunedin cohort in which DunedinPoAm was developed and the Understanding Society, NAS, and E-Risk cohorts and CALERIE trial sample in which it was tested were mostly of white European descent. Follow-up of DunedinPoAm in more diverse samples is needed to establish cross-population validity. Finally, because methylation data are not yet available from CALERIE follow-up assessments, we could not test if intervention modified DunedinPoAm at outcome. Ultimately, to establish DunedinPoAm as a surrogate endpoint for healthspan, it will be necessary to establish not only robust association with healthy lifespan phenotypes and modifiability by intervention, but also the extent to which changes in DunedinPoAm induced by intervention correspond to changes in healthy-lifespan phenotypes (*Prentice, 1989*).

Within the bounds of these limitations, our analysis establishes proof-of-concept for DunedinPoAm as a single-time-point measure that quantifies Pace of Aging from a blood test. It can be implemented in Illumina 450 k and EPIC array data, making it immediately available for testing in a wide range of existing datasets as a complement to existing methylation measures of aging. Critically, DunedinPoAm offers a unique measurement for intervention trials and natural experiment studies investigating how the rate of aging may be changed by behavioral or drug therapy, or by environmental modification. DunedinPoAm may be especially valuable to studies that collect data outside of clinical settings and lack blood chemistry, hematology, and other data needed to measure aging-related changes to physiology.

## Materials and methods

### Data sources

Data were used from five studies: the Dunedin Study, the Understanding Society Study, the Normative Aging Study (NAS), the Environmental Risk (E-Risk) Longitudinal Twin Study, and the CALERIE Trial. The five datasets and measures analyzed within each of them are described below.

**The Dunedin Study** is a longitudinal investigation of health and behavior in a complete birth cohort. Study members (N = 1,037; 91% of eligible births; 52% male) were all individuals born between April 1972 and March 1973 in Dunedin, New Zealand (NZ), who were eligible based on residence in the province and who participated in the first assessment at age 3. The cohort represents the full range of socioeconomic status on NZ's South Island and matches the NZ National Health and Nutrition Survey on key health indicators (e.g., BMI, smoking, GP visits) (*Poulton et al., 2015*). The cohort is primarily white (93%) (*Poulton et al., 2015*). Assessments were carried out at birth and ages 3, 5, 7, 9, 11, 13, 15, 18, 21, 26, 32, 38 and, most recently, 45 years, when 94% of the 997 study members still alive took part. At each assessment, each study member is brought to the research unit for a full day of interviews and examinations. Study data may be accessed through agreement with the Study investigators (https://moffittcaspi.trinity.duke.edu/research-topics/dunedin). Dunedin Study measures of physical and cognitive functioning and subjective signs of aging are described in detail in *Supplementary file 1H*.

**Understanding Society** is an ongoing panel study of the United Kingdom population (https://www.understandingsociety.ac.uk/). During 2010–12, participants were invited to take part in a nurse's exam involving a blood draw. Of the roughly 20,000 participants who provided clinical data in this exam, methylation data have been generated for just under 1200. We analyzed data from 1175 participants with available methylation and blood chemistry data. Documentation of the methylation (*University of Essex, 2012*) and blood chemistry (*University of Essex, 2017*) data resource

is available online (https://www.understandingsociety.ac.uk/sites/default/files/downloads/documentation/health/user-guides/7251-UnderstandingSociety-Biomarker-UserGuide-2014.pdf).

Klemera-Doubal method (KDM) Biological Age. We measured KDM Biological age from blood chemistry, systolic blood pressure, and lung-function data using the algorithm proposed by *Klemera and Doubal (2006)* trained in data from the NHANES following the method originally described by *Levine (2013)* and using the dataset compiled by Hastings (*Hastings et al., 2019*). We included 8 of Levine's original 10 biomarkers in the algorithm: albumin, alkaline phosphatase (log), blood urea nitrogen, creatinine (log), C-reactive protein (log), HbA1C, systolic blood pressure, and forced expiratory volume in 1 s ($FEV_1$). We omitted total cholesterol because of evidence this biomarker shows different directions of association with aging in younger and older adults (*Arbeev et al., 2016*). Cytomegalovirus optical density was not available in the Understanding Society database.

Self Rated Health. Understanding Society participants rated their health as excellent, very-good, good, fair, or poor. We standardized this measure to have Mean = 0, Standard Deviation = 1 for analysis.

**The Normative Aging Study** (**NAS**) is an ongoing longitudinal study on aging established by the US Department of Veterans Affairs in 1963. Details of the study have been published previously (*Bell et al., 1972*). Briefly, the NAS is a closed cohort of 2280 male veterans from the Greater Boston area enrolled after an initial health screening to determine that they were free of known chronic medical conditions. Participants have been re-evaluated every 3–5 years on a continuous rolling basis using detailed on-site physical examinations and questionnaires. DNA from blood samples was collected from 771 participants beginning in 1999. We analyzed blood DNA methylation data from up to four repeated assessments conducted through 2013 (*Gao et al., 2019b*; *Panni et al., 2016*). Of the 771 participants with DNA methylation data, n = 536 (46%) had data from two repeated assessments and n = 178 (23%) had data from three or four repeated assessments. We restricted the current analysis to participants with at least one DNA methylation data point. The NAS was approved by the Department of Veterans Affairs Boston Healthcare System and written informed consent was obtained from each subject before participation.

Mortality. Regular mailings to study participants have been used to acquire vital-status information and official death certificates were obtained from the appropriate state health department to be reviewed by a physician. Participant deaths are routinely updated by the research team and the last available update was on 31 December 2013. During follow-up, n = 355 (46%) of the 771 NAS participants died.

Chronic Disease Morbidity. We measured chronic disease morbidity from participants medical histories and prior diagnoses (*Gao et al., 2019a*; *Gao et al., 2019c*; *Lepeule et al., 2018*; *Nyhan et al., 2018*). We counted the number of chronic diseases to compose an ordinal index with categories of 0, 1, 2, 3, or 4+ of the following comorbidities: hypertension, type-2 diabetes, cardiovascular disease, chronic obstructive pulmonary disease, chronic kidney disease, and cancer.

**The Environmental Risk (E-Risk) Longitudinal Twin Study** tracks the development of a birth cohort of 2,232 British participants. The sample was drawn from a larger birth register of twins born in England and Wales in 1994–1995. Full details about the sample are reported elsewhere (*Moffitt and E-Risk Study Team, 2002*). Briefly, the E-Risk sample was constructed in 1999–2000, when 1116 families (93% of those eligible) with same-sex 5-year-old twins participated in home-visit assessments. This sample comprised 56% monozygotic (MZ) and 44% dizygotic (DZ) twin pairs; sex was evenly distributed within zygosity (49% male). Families were recruited to represent the UK population of families with newborns in the 1990 s, on the basis of residential location throughout England and Wales and mother's age. Teenaged mothers with twins were over-selected to replace high-risk families who were selectively lost to the register through non-response. Older mothers having twins via assisted reproduction were under-selected to avoid an excess of well-educated older mothers. The study sample represents the full range of socioeconomic conditions in the UK, as reflected in the families' distribution on a neighborhood-level socioeconomic index (ACORN [A Classification of Residential Neighborhoods], developed by CACI Inc for commercial use): 25.6% of E-Risk families lived in 'wealthy achiever' neighborhoods compared to 25.3% nationwide; 5.3% vs. 11.6% lived in 'urban prosperity' neighborhoods; 29.6% vs. 26.9% lived in 'comfortably off' neighborhoods; 13.4% vs. 13.9% lived in 'moderate means' neighborhoods, and 26.1% vs. 20.7% lived in 'hard-pressed' neighborhoods. E-Risk underrepresents 'urban prosperity' neighborhoods because such households are likely to be childless.

Home-visits assessments took place when participants were aged 5, 7, 10, 12 and, most recently, 18 years, when 93% of the participants took part. At ages 5, 7, 10, and 12 years, assessments were carried out with participants as well as their mothers (or primary caretakers); the home visit at age 18 included interviews only with participants. Each twin was assessed by a different interviewer. These data are supplemented by searches of official records and by questionnaires that are mailed, as developmentally appropriate, to teachers, and co-informants nominated by participants themselves. The Joint South London and Maudsley and the Institute of Psychiatry Research Ethics Committee approved each phase of the study. Parents gave informed consent and twins gave assent between 5–12 years and then informed consent at age 18. Study data may be accessed through agreement with the Study investigators (https://moffittcaspi.trinity.duke.edu/research-topics/erisk).

Childhood Socioeconomic Status (SES). Childhood SES was defined through a standardized composite of parental income, education, and occupation (*Trzesniewski et al., 2006*). The three SES indicators were highly correlated (r = 0.57–0.67) and loaded significantly onto one factor. The population-wide distribution of the resulting factor was divided in tertiles for analyses.

Childhood Victimization. As previously described (*Danese et al., 2017*), we assessed exposure to six types of childhood victimization between birth to age 12: exposure to domestic violence between the mother and her partner, frequent bullying by peers, physical and sexual harm by an adult, and neglect.

**The CALERIE Trial** is described in detail elsewhere (*Ravussin et al., 2015*). Briefly, N = 220 normal-weight (22.0 $\leq$ BMI < 28 kg/m$^2$) participants (70% female, 77% white) aged 21–50 years at baseline were randomized to caloric restriction or ad libitum conditions with a 2:1 ratio (n = 145 to caloric restriction, n = 75 to ad libitum). 'Ad libitum' (normal) caloric intake was determined from two consecutive 14 day assessments of total daily energy expenditure using doubly labeled water (*Redman et al., 2014*). Average percent caloric restriction over six-month intervals was retrospectively calculated by the intake-balance method with simultaneous measurements of total daily energy expenditure using doubly labeled water and changes in body composition (*Racette et al., 2012*; *Wong et al., 2014*). Over the course of the trial, participants in the caloric-restriction arm averaged 12% reduction in caloric intake (about half the prescribed reduction). Participants in the ad libitum condition reduced caloric intake by <2% (*Ravussin et al., 2015*). CALERIE data are available at https://calerie.duke.edu/samples-data-access-and-analysis.

Klemera-Doubal method (KDM) Biological Age. KDM Biological age was measured according to the procedure described in our previous article (*Belsky et al., 2018a*). Briefly, we computed KDM Biological Age from CALERIE blood chemistry and blood pressure data using the algorithm proposed by *Klemera and Doubal (2006)* trained in data from the NHANES following the method originally described by *Levine (2013)* and NHANES data from years matched to the timing of the CALERIE Trial. We included 8 of Levine's original 10 biomarkers in the algorithm: albumin, alkaline phosphatase (log), blood urea nitrogen, creatinine (log), C-reactive protein (log), HbA1C, systolic blood pressure, and total cholesterol. Cytomegalovirus optical density and lung function were not measured in CALERIE. We supplemented the algorithm with data on uric acid and white blood cell count.

## DNA methylation data

DNA methylation was measured from Illumina 450 k Arrays in the Dunedin Study, NAS, and E-Risk Study and from Illumina EPIC 850 k Arrays in the Understanding Society study and the CALERIE Trial. DNA was derived from whole blood samples in all studies. Dunedin Study blood draws were conducted at the cohort's age-38 assessment during 2010–12. Understanding Society blood draws were conducted in 2012. NAS blood draws were conducted during 1999–2013. E-Risk blood draws were conducted at the cohort's age-18 assessment during 2012–13. CALERIE blood draws were conducted at the trial baseline assessment in 2007. Dunedin and CALERIE methylation assays were run by the Molecular Genomics Shared Resource at Duke Molecular Physiology Institute, Duke University (USA). Understanding Society and E-Risk assays were run by the Complex Disease Epigenetics Group at the University of Exeter Medical School (UK) (www.epigenomicslab.com). NAS methylation assays were run by the Genome Research Core of the University of Illinois at Chicago. Processing protocols for the methylation data from all studies have been described previously (*Dai et al., 2017*; *Hannon et al., 2018*; *Marzi et al., 2018*; *Panni et al., 2016*). (CALERIE data were processed according to the same protocols used for the Dunedin Study.)

## Methylation clocks

We computed the methylation clocks proposed by Horvath, Hannum, and Levine using the methylation data provided by the individual studies and published algorithms (*Hannum et al., 2013*; *Horvath, 2013*; *Levine et al., 2018*).

## DunedinPoAm

The Dunedin Pace of Aging methylation algorithm (DunedinPoAm) was developed using elastic-net regression analysis carried out in the Dunedin Study, as described in detail in the Results. The criterion variable was Pace of Aging. Development of the Pace of Aging is described in detail elsewhere (*Belsky et al., 2015*). Briefly, we conducted mixed-effects growth modeling of longitudinal change in 18 biomarkers measuring integrity of the cardiovascular, metabolic, renal, hepatic, pulmonary, periodontal, and immune systems. Biomarkers were measured at the age 26, 32, and 38 assessments: in order of listing in Figure 3 of *Belsky et al. (2015)* glycated hemoglobin, cardiorespiratory fitness, waist-hip ratio, $FEV_1/FVC$ ratio, $FEV_1$, mean arterial pressure, body mass index, leukocyte telomere length, creatinine clearance, blood urea nitrogen, lipoprotein (a), triglycerides, gum health, total cholesterol, white blood cell count, high-sensitivity C-reactive protein, HDL cholesterol, ApoB100/ApoA1 ratio. For each biomarker, we estimated random slopes quantifying each participant's own rate of change in that biomarker. We then composited slopes across the 18 biomarkers to calculate a participant's Pace of Aging. Pace of Aging was scaled in units representing the mean trend in the cohort, that is the average physiological change occurring during one calendar year (N = 954, M = 1, SD = 0.38). Of the N = 819 Dunedin Study members with methylation data at age 38, N = 810 had measured Pace of Aging (M = 0.98, SD = 0.09). This group formed the analysis sample to develop DunedinPoAm.

To compute DunedinPoAm in the Understanding Society, NAS, E-Risk, and CALERIE Trial datasets, we applied the scoring algorithm estimated from elastic net regression in the Dunedin Study. CpG weights for the scoring algorithm are provided in *Supplementary file 1A*. R code to implement the scoring algorithm in data from Illumina 450 k and EPIC arrays is provided at https://github.com/danbelsky/DunedinPoAm38 .

## Bootstrap repetition analysis to estimate out-of-sample correlation between methylation Pace of Aging (mPoA) measures and longitudinal Pace of Aging

The Dunedin Study is the only dataset to include measured 12 year longitudinal Pace of Aging. To estimate the out-of-sample correlation between mPoA and the original Pace of Aging measure, we conducted 90/10 crossfold validation analysis. We randomly selected 90% of the cohort to serve as the 'training' sample in which the mPoA algorithm was developed. We used the remaining 10% to form a 'test' sample to estimate the correlation between mPoA and Pace of Aging. We repeated this analysis across 100 bootstrap repetitions. In each repetition, we randomly sampled 90% of the cohort to use in the training analysis and reserved the remaining 10% for testing.

The mPoA algorithms developed across the 100 bootstrap repetitions included different sets of CpGs (range of 21–209 CpGs selected, M = 54, SD = 27 CpGs). However, the resulting algorithms were highly correlated (mean pairwise r = 0.90, SD = 0.14). The average correlation between the 90%-trained mPoA and longitudinal Pace of Aging in the 10% test samples was (r = 0.33, SD = 0.10). Details are reported in *Figure 1—figure supplement 2*.

## Statistical analysis

We conducted analysis of Dunedin, Understanding Society, NAS, E-Risk, and CALERIE data using regression models. We analyzed continuous outcome data using linear regression. We analyzed count outcome data using Poisson regression. We analyzed time-to-event outcome data using Cox proportional hazard regression. For analysis of repeated-measures longitudinal DNA methylation data in the NAS, we used generalized estimating equations to account for non-independence of repeated observations of individuals (*Ballinger, 2004*), following the method in previous analysis of those data (*Gao et al., 2018*), and econometric fixed-effects regression (*Wooldridge, 2012*) to test within-person change over time. For analysis in E-Risk, which include data on twin siblings, we clustered standard errors at the family level to account for non-independence of data. For analysis of

longitudinal change in clinical-biomarker biological age in CALERIE, we used mixed-effects growth models (*Singer and Willett, 2003*) following the method in our original analysis of those data (*Belsky et al., 2018a*). For regression analysis, methylation measures were adjusted for batch effects by regressing the measure on batch controls and predicting residual values. Dunedin Study, Understanding Society, E-Risk, and CALERIE analyses included covariate adjustment for sex (the NAS included only men). Understanding Society, NAS, and CALERIE analyses included covariate adjustment for chronological age. (Dunedin and E-Risk are birth-cohort studies and participants are all the same chronological age.) Sensitivity analyses testing covariate adjustment for estimated leukocyte distributions and smoking are reported in *Supplementary file 1C-G*.

## Sensitivity analyses

We tested sensitivity of methylation Pace of Aging (mPoA) measures to alternative methods of normalizing DNA methylation data. We normalized data using the 'methylumi' and 'minfi' packages and computed correlations between mPoA measures derived from these two datasets. The correlation was r = 0.94.

The elastic net model selected 46 CpGs to compose the mPoA. One of these CpGs, cg11897887, has been identified as an mQTL (*Volkov et al., 2016*). To evaluate sensitivity of results to the exclusion of this CpGs, we computed a version of the mPoA excluding this CpG and repeated analysis. This version of the score was correlated with the full mPoA at r = 1. Results were the same in analyses with both versions (available from the authors upon request).

Another CpG selected in the elastic net, cg05575921, is located within the gene *AHRR*, previously identified as a methylation site modified by tobacco exposure and associated with lung cancer and other chronic disease, for example (*Fasanelli et al., 2015*; *Reynolds et al., 2015*). We tested sensitivity of results to the exclusion of this probe using the method described above. This version of the score was correlated with the full mPoA at r = 0.94. Again, results were the same in analyses with both versions (available from the authors upon request).

## Code for analysis

Code used for analysis and to prepare figures is accessible via https://github.com/danbelsky/DunedinPoAm_eLife2020.git (*Belsky, 2020*; copy archived at https://github.com/elifesciences-publications/DunedinPoAm_eLife2020).

Code to calculate DunedinPoAm from Illumina 450 k or Epic Array Data. R code is available at https://github.com/danbelsky/DunedinPoAm38.git.

# Acknowledgements

This research was supported by US-National Institute on Aging grants AG032282 and UK Medical Research Council grant MR/P005918/1. The Dunedin Multidisciplinary Health and Development Research Unit is supported by the New Zealand Health Research Council Programme Grant (16-604), and the New Zealand Ministry of Business, Innovation and Employment (MBIE). We thank the Dunedin Study members, Unit research staff, and Study founder Phil Silva.

Understanding Society data come from The UK Household Longitudinal Study, which is led by the Institute for Social and Economic Research at the University of Essex and funded by the Economic and Social Research Council (ES/M008592/1). The data were collected by NatCen and the genome wide scan data were analysed by the Wellcome Trust Sanger Institute. Information on how to access the data can be found on the Understanding Society website https://www.understanding-society.ac.uk/. Data governance was provided by the METADAC data access committee, funded by ESRC, Wellcome, and MRC (2015–2018: MR/N01104X/1; 2018–2020: ES/S008349/1)

The Normative Aging Study is supported by the National Institute of Environmental Health Sciences (grants P30ES009089, R01ES021733, R01ES025225, and R01ES027747). The VA Normative Aging Study is supported by the Cooperative Studies Program/Epidemiology Research and Information Center of the U.S. Department of Veterans Affairs and is a component of the Massachusetts Veterans Epidemiology Research and Information Center, Boston, Massachusetts.

The E-Risk Study is supported by the UK Medical Research Council (grant G1002190), the US National Institute of Child Health and Development (grant HD077482), and the Jacobs Foundation. The generation of DNA methylation data was supported by the American Asthma Foundation.

This investigation was made possible in part through use of the CALERIE data repository and was supported in part by U24AG047121 and R01AG061378.

This work used a high-performance computing facility partially supported by grant 2016-IDG-1013 (HARDAC+: Reproducible HPC for Next-generation Genomics") from the North Carolina Biotechnology Center.

DWB was additionally supported by US National Institute on Aging grant R21AG054846, the CIFAR Child Brain Development Network, and the Jacobs Foundation.

## Additional information

### Funding

| Funder | Grant reference number | Author |
| --- | --- | --- |
| Medical Research Council | MR/P005918/1 | Terrie E Moffitt |
| Medical Research Council | G1002190 | Terrie E Moffitt |
| National Institute on Aging | R01AG032282 | Terrie E Moffitt |
| National Institute on Aging | U24AG047121 | William E Kraus |
| National Institute on Aging | R01AG061378 | Daniel W W Belsky |
| National Institute on Aging | R21AG054846 | Daniel W W Belsky |
| National Institute of Child Health and Development | R01HD077482 | Avshalom Caspi |
| CIFAR | CBD Network Fellowship | Daniel W Belsky |
| Jacobs Foundation | | Daniel W Belsky Terrie E Moffitt Avshalom Caspi |

The funders had no role in study design, data collection and interpretation, or the decision to submit the work for publication.

### Author contributions

Daniel W Belsky, Conceptualization, Resources, Formal analysis, Funding acquisition, Writing - original draft, Writing - review and editing; Avshalom Caspi, Terrie E Moffitt, Conceptualization, Resources, Funding acquisition, Writing - original draft, Writing - review and editing; Louise Arseneault, Jonathan Mill, Richie Poulton, Resources, Funding acquisition, Writing - review and editing; Andrea Baccarelli, Resources, Writing - review and editing; David L Corcoran, Joseph A Prinz, Data curation, Formal analysis, Writing - review and editing; Xu Gao, Formal analysis, Writing - review and editing; Eiliss Hannon, Hona Lee Harrington, Karen Sugden, Data curation, Writing - review and editing; Line JH Rasmussen, Renate Houts, Kim Huffman, William E Kraus, Carl F Pieper, Benjamin S Williams, Writing - review and editing; Dayoon Kwon, Formal analysis; Joel Schwartz, Pantel Vokonas, Resources

### Author ORCIDs

Daniel W Belsky https://orcid.org/0000-0001-5463-2212
Line JH Rasmussen https://orcid.org/0000-0001-6613-2469

### Ethics

Human subjects: Research participants provided informed consent for participation in the included studies. Approval for this study was provided by the institutional review boards of the New Zealand Ministry of Health Health and Disability Ethics Committee (17/STH/25/AM05), Duke University (protocols 1604 and 0414B0360), Duke University Medical Center (00078391) and Columbia University Irving Medical Center (protocols AAAS2948 and AAAQ8739).

Decision letter and Author response
Decision letter https://doi.org/10.7554/eLife.54870.sa1
Author response https://doi.org/10.7554/eLife.54870.sa2

# Additional files

### Supplementary files

• Supplementary file 1. The supplementary file contains supplemental tables and figures. Tables include (1) a list of the CpGs included in the DunedinPoAm algorithm and their associated weights, (2-7) results from regression models reported in the main text, and (8) a description of the functional test measures from the Dunedin Study. Supplementary figures show (1) analysis of the Dunedin Study to develop Dunedin PoAm and (1-6) effect-sizes for associations of DunedinPoAm and the epigenetic clocks porposed by Horvath, Hannum et al., and Levine et al. with outcome measures.

• Transparent reporting form

### Data availability

Datasets are available from the data owners. Data from the Dunedin and E-Risk Study can be accessed through agreement with the Study investigators. Instructions are available at https://sites.google.com/site/moffittcaspiprojects/. The data access application form can be downloaded here: https://sites.google.com/site/moffittcaspiprojects/forms-for-new-projects/concept-paper-template. Data from the Understanding Society Study is available through METADAC at https://www.metadac.ac.uk/ukhls/. All details are on the Metadac website (https://www.metadac.ac.uk/data-access-through-metadac/). The data access application form can be found here https://www.metadac.ac.uk/files/2019/02/v2.41-UKHLS-METADAC-application-form-2019-2hak8bv.docx. Data from the Normative Aging Study were obtained from the Study investigators. Data are accessible through dbGaP, accession phs000853.v1.p1. CALERIE Data are available for download from the CALERIE Biorepository at https://calerie.duke.edu/. Details are on the website here: https://calerie.duke.edu/samples-data-access-and-analysis.

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
