## [Decision Letter]

**Acceptance summary:**

The paper is a nice contribution to the research field of epigenetics and aging with thorough epidemiological analyses and cohort data included. We are delighted to include it in the special issue on Aging, Geroscience and Longevity in *eLife*.

**Decision letter after peer review:**

[Editors’ note: the authors submitted for reconsideration following the decision after peer review. What follows is the decision letter after the first round of review.]

Thank you for submitting your work entitled "Quantification of the pace of biological aging in humans through a blood test: a DNA methylation algorithm" for consideration by *eLife*. Your article has been reviewed by four peer reviewers, including Sara Hägg as the Reviewing Editor and Reviewer #1, and the evaluation has been overseen by a Senior Editor. The following individuals involved in review of your submission have agreed to reveal their identity: Jamie Justice (Reviewer #3); Matthew Suderman (Reviewer #4).

Our decision has been reached after consultation between the reviewers. We regret to inform you that your work will not be considered for publication in *eLife* in its current form. However, there is an opportunity for resubmission of a revised paper as a new submission, which will go to the same reviewers and the same editors.

*Reviewer #1:*I think this is a nice manuscript and I only have one major comment, and that is that the mPoA should be validated in longitudinal methylation data in an aging cohort. I think this is critical.

Reviewer #2:

This is an interesting study that aims to develop a new "biological age" marker, termed mPoA, following a similar methodology to Horvath's famous epigenetic clock. However, it is conceptually more similar to Levine's methylation phenotypic age, being based on a two-step process involving clinical markers. The epigenetic measure presented here is novel since it attempts to quantify in a single measurement and single assay "Pace of aging", which was originally developed in a New Zealand cohort of young adults, using standard clinical markers. An impressive range of studies are then used for partial validation of the measure. However, I would not say it a scientific breakthrough of the highest significance since it provides yet another epigenetic age marker, or which its meaning is not totally clear, without providing evidence of improved performance in predicting health/mortality than currently available measures.

1) While the associations shown in a range of contexts is impressive, it is a shame that a collaboration was not sought with an older cohort to demonstrate performance in predicting mortality/age-related morbidity. Since it is not conceptually clear what an epigenetic predictor of rate of change of clinical parameters between 26 and 38 years means in other age groups and contexts, I think a pre-requisite of a new biological ageing marker is demonstration of utility in predicting health/lifespan.

2) The authors should discuss why an epigenetic marker with a fairly moderate correlation with original PoA performs in many cases better than the original PoA in predicting subsequent functional ability. A current concern with epigenetic age predictors (e.g. https://www.biorxiv.org/content/10.1101/327890v2.full) is confounding by cell composition. This may provide an explanation. The authors should attempt to address this in sensitivity analyses (one possible approach could be including cell composition as a fixed variable in the elastic net model, and then regressing this out for the final measure)

3) The authors should be commended for publishing the models predictors in full (they could also provide a code for other researcher to easily compute the measure in full) however some discussion on the model predictors is needed to understand what epigenetic changes are driving the model and if any biological pathways are over-represented.

4) It would be useful to provide comparisons with the other biological age measures in each of the analyses – i.e. is mPoA more sensitive to adversity in eRisk than hovarth, levine etc. or does it predict difference in subsequent aging in the RCT better than KDM BA or Levine etc or even original PoA

5) The Abstract is a bit light on numbers – please add the important details such a number of participants, effects sizes, statistical significance estimates for the main results.*Reviewer #3:*Study synopsis. The authors train the methylation Pace of Aging (mPoA) predictor in adults aged 38 enrolled in the Dunedin study, and use it to predict subsequent change in the physiologic parameters used to calculate PoA in the same cohort (age 38-45). They next perform analyses of mPoA (measured at one timepoint) in other cohorts and clinical study types to validate mPoA and explore associations with existing methylation and composite biomarkers. Validation studies include Understanding Society Study (cross-sectional, mixed age sample with existing methylation data), E-risk Study (cross-sectional; adolescent / young adults with distinct socio-environmental risk factors), and CALERIE (clinical trial, age-similar to Dunedin training cohort). Authors are commended on an excellent contribution; concerns and comments are relatively minor and should not pose a barrier to publication.

In general, this is a meritorious and well written addition to a progression of work on biomarkers of biological age in (early) middle-aged adults. The authors have previously evaluated multivariable composite biomarkers in the Dunedin longitudinal cohort and calculated the Pace of Aging (PoA). In the present study the investigators use this composite measure of change as the foundation to train a DNA-wide methylation clock tuned to PoA. In doing so, the investigators have created a first-of-its kind methylation biomarker (mPoA) to designed specifically to predict change over time rather than a static, single-timepoint measure (e.g. clocks tuned to chronologic age or phenotype). A methylation biomarker tuned to detect change is a significant contribution that could have utility as a biomarker in clinical trials designed to prevent or delay declines in health and physiologic function with age.

1) The findings in CALERIE trial are perhaps the most exciting results: mPoA measured at baseline discriminated change in the multivariable composite KDM Biological Age. It is also suggestive of potential to discriminate between treatment arms of a robust intervention on the biology of aging (caloric restriction), though this needs to be followed up with longitudinal assessment. These noteworthy findings should be emphasized. Authors suggested to restructure presentation of results. In the Abstract authors state that mPoA "was disrupted by caloric restriction in the CALERIE trial." mPoA was only measured at baseline not tracked longitudinally for between group effects on change in mPoA as implied by statement. Please revise.

2) mPoA measured at single timepoint.

• That mPoA is only measured at baseline in CALERIE and at only one timepoint in the validation studies dampens enthusiasm. While the findings in the CALERIE study (Figure 4) are a substantial advancement, biomarkers with repeat measurement are critically needed. Biomarkers with test-retest reliability, sensitivity to change, and predict clinical trial endpoints (e.g. disability onset or incident disease vs. KDM change) are essential to progress. This is noted in limitations. Given iterative nature of research the reviewer trusts that this will be addressed in a future study – hopefully a study repeating mPoA at age 45 in the Dunedin cohort, or at Year 1 in the CALERIE trial when/if additional methylation data become available.

• While longitudinal change in mPoA cannot be included in this manuscript, two minor suggestions for context:

2A) The TRIIM clinical study should be referenced (a single arm clinical study in 9 men; Fahy et al., 2019). It is currently the only published report of methylation biomarker performance over time in an intervention study.

2B) The authors correctly state that "measurements of the rate of biological aging are needed to serve as surrogate endpoints in trials…" This is important, and additional context is needed. For example, criteria to validate surrogate endpoints would be pertinent- e.g. Prentice criteria: biomarker to be considered as a surrogate, it should a) be correlated with clinical endpoint; b) capture net effect of intervention on clinical endpoint. This will provide context and forecast next steps in logical progression of biomarker development (e.g. longitudinal assessment and prediction of clinical trial outcome).

3) Dunedin study.

• Many of the physical performance tests in Dunedin Study are likely to have ceiling effects and limited change over time would be expected given the ages and health status of the study population. That the investigators find anything useful in change over time to train the methylation clock is a wonder (reviewer anticipated "flat" findings like those for grip strength). Given the history of work with this cohort revision is not requested. In future it would be interesting to see how mPoA would perform if the algorithm were trained in a more diverse population with greater observable deficits in physiologic function over time.

3A) The comparison of adult (age 45) to juvenile (age 7-13) Wechsler Intelligence Scales as an index of 'cognitive decline' is puzzling. Please provide validation and justification.

3B) Please provide references on validation work for the facial aging in the text or supplement.

*Reviewer #4:*This very well-written manuscript presents a new method for estimating the rate of biological aging from blood DNA methylation (mPOA), a linear model based on blood DNA methylation of an published model that was derived from 18 biomarkers called Pace of Aging (POA). mPOA differs from other DNA methylation models of biological aging (Horvath, Hannum, Levine) by ignoring chronological age. mPOA appears to be a truly novel and valuable model of aging:

- mPOA predicted functional decline and subjective age-related changes in mid-life, 7 years in advance – apparently better than the original POA model on which it is defined

- mPOA appears to have a stronger association with Klemera and Doubal Biological Age (similar to POA) and likely to POA (although this is never tested) than other DNA methylation-based models

- mPOA is associated with childhood socioeconomic status and victimization at age 18

- mPOA *may* show a protective effect after only two years of calorie restriction

This manuscript would be a valuable addition to the aging literature.

A major limitation of previous DNA-methylation based models of aging is the difficulty of arriving at some kind of biological interpretation of their behavior because they are derived from chronological age. mPOA is, by contrast, derived directly from "blood- and organ-system-functional biomarkers". As a result, it should be possible to investigate the relative importance of each of these biomarkers. It could help to explain how and why mPOA differs from POA and where and when it might be more or less informative. At the very least the manuscript could include a table and/or figure comparing the associations of mPOA and POA with the 18 biomarkers.

There is a distinct lack of negative/null findings in the paper suggesting that such findings may have been omitted. For example, mPOA appears to be associated with all measures of aging (except for grip strength, the only measure that apparently increases with age), and Figure 2—figure supplement 1 suggests that mPOA performs better than POA for all (except grip strength). Some functioning and aging measures used to investigate POA are not included in the present study (see e.g. Belsky et al., 2015). Why are they omitted? If mPOA really is better than POA, this should at least draw some discussion if not some follow-up analysis. It suggests that DNA methylation captures some informative biological variation lacking in the 18 underlying biomarkers, perhaps cell count variation, smoking behavior, or age?

Given the significant amount of published work related to the 'epigenetic clocks', it is critical to show that any new DNA methylation-based model of aging is not only different from previous models but superior in important ways. The manuscript only compares mPOA to other models in terms of associations with KDM Biological Age. In this comparison, mPOA is clearly superior to Horvath and Hannum, but for Levine the comparison is not so clear. It should be easy to derive Levine in the Dunedin data and test associations with functional decline and subjective aging (in fact, haven't Horvath and Hannum have already been derived in Dunedin? – see Belsky et al., 2018).

Smoking status appears central to mPOA – of the 46 CpG sites on which mPOA is based, at least 12 have been previously linked with smoking. This is explored only for the comparison with the 'epigenetic clocks' in Understanding Society and for assessing associations with SES and victimization in E-Risk. To what extent does smoking status influence the association between mPOA and POA? To what extent does smoking status influence associations/predictions of functional decline and subjective aging in Dunedin?

---

## [Author Response]

[Editors’ note: The authors appealed the original decision. What follows is the authors’ response to the first round of review.]

Reviewer #1:I think this is a nice manuscript and I only have one major comment, and that is that the mPoA should be validated in longitudinal methylation data in an aging cohort. I think this is critical.

Thank you for the positive comments. We have added analysis of longitudinal, repeated-measures DNA methylation data from the Normative Aging Study (Discussion, second paragraph). These data now validate that our measure, renamed “DunedinPoAm” in our revised submission, increases with advancing chronological age, consistent with demographic observations of accelerating mortality risk at older ages.

Reviewer #2:This is an interesting study that aims to develop a new "biological age" marker, termed mPoA, following a similar methodology to Horvath's famous epigenetic clock. However, it is conceptually more similar to Levine's methylation phenotypic age, being based on a two-step process involving clinical markers. The epigenetic measure presented here is novel since it attempts to quantify in a single measurement and single assay "Pace of aging", which was originally developed in a New Zealand cohort of young adults, using standard clinical markers. An impressive range of studies are then used for partial validation of the measure. However, I would not say it a scientific breakthrough of the highest significance since it provides yet another epigenetic age marker, or which its meaning is not totally clear, without providing evidence of improved performance in predicting health/mortality than currently available measures.

Thank you for the opportunity to clarify. Our measure, renamed “Dunedin PoAm” in this revision, is qualitatively different from published epigenetic clocks, including the Horvath clock. We have added text to clarify this point, quoted below, and also a new Figure 1 to better clarify how DunedinPoAm was developed.

“DunedinPoAm is qualitatively different from previously published DNA methylation measures of aging that were developed by comparing older individuals to younger ones. […] DunedinPoAm is designed to function as a speedometer, recording how fast the subject is aging.”

In addition, we provide novel evidence of the improved prediction of health and mortality of DunedinPoAm in the subsection “DunedinPoAm was associated with chronic disease morbidity and increased risk of mortality among older men in the Normative Aging Study (NAS)” and Supplementary file 1E.

1) While the associations shown in a range of contexts is impressive, it is a shame that a collaboration was not sought with an older cohort to demonstrate performance in predicting mortality/age-related morbidity. Since it is not conceptually clear what an epigenetic predictor of rate of change of clinical parameters between 26 and 38 years means in other age groups and contexts, I think a pre-requisite of a new biological ageing marker is demonstration of utility in predicting health/lifespan.

Thank you for this suggestion. We have added analysis of an older-adult cohort, the Normative Aging Study. We report analysis of mortality, of incident and prevalent chronic disease, and of within-person change over time. DunedinPoAm is associated with mortality, future incidence of chronic disease, and prevalent chronic disease. Results are reported in the subsection “DunedinPoAm was associated with chronic disease morbidity and increased risk of mortality among older men in the Normative Aging Study (NAS)” Figure 4, and Supplementary file 1E.

2) The authors should discuss why an epigenetic marker with a fairly moderate correlation with original PoA performs in many cases better than the original PoA in predicting subsequent functional ability. A current concern with epigenetic age predictors (e.g. https://www.biorxiv.org/content/10.1101/327890v2.full) is confounding by cell composition. This may provide an explanation. The authors should attempt to address this in sensitivity analyses (one possible approach could be including cell composition as a fixed variable in the elastic net model, and then regressing this out for the final measure)

Done. We now comment on the difference in effect-sizes between the original Pace of Aging measure and our new measure, DunedinPoAm, in the Discussion:

“Effect-sizes were equal to or greater than those for the 18-biomarker 3-time point measure of Pace of Aging. This result suggests that the DNA methylation elastic-net regression used to develop DunedinPoAm may have distilled the aging signal from the original Pace of Aging measure and excluded some noise.”

We report analysis including covariate adjustment for estimated leukocyte distributions using Houseman’s method (Houseman et al., 2012) in Supplementary file 1C-G. Statistical adjustment for estimated cell counts did not change results.

3) The authors should be commended for publishing the models predictors in full (they could also provide a code for other researcher to easily compute the measure in full).

Thank you for this suggestion. We have now added a “projector” package to the Materials and methods subsection “Code for analysis”, that will enable readers to compute DunedinPoAm in their own methylation data.

However some discussion on the model predictors is needed to understand what epigenetic changes are driving the model and if any biological pathways are over-represented.

Thank you for the opportunity to clarify this issue. The elastic-net regression method we used to develop DunedinPoAm is agnostic to the biology measured by the DNA methylation marks studied. It produces a statistical summary of variance in the criterion measure, in our analysis, Pace of Aging. That statistical summary may or may not prioritize DNA methylation marks in or near genes relevant to the aging process.

We report the nearest gene to each CpG included in the algorithm in Supplementary file 1A. To address the reviewers concern, we uploaded this list to pantherdb.org to conduct genes-setenrichment analysis. There were no GO biological processes or PANTHER pathways enriched in this list at an FDR-corrected level of statistical significance. This means that DunedinPoAm is a statistical summary of variation in the multiple-biomarker measure on which it was trained, Pace of Aging. We have not included output from these analyses with our manuscript because they are easily reproduced using the gene list in Supplementary file 1A.

4) It would be useful to provide comparisons with the other biological age measures in each of the analyses -i.e. is mPoA more sensitive to adversity in eRisk than hovarth, levine etc or does it predict difference in subsequent aging in the RCT better than KDM BA or Levine etc or even original PoA

Done. We now comment on this comparison at the end of each section of the Results and include tables and figures reporting effect-sizes for DunedinPoAm and the three epigenetic clocks in each of the analyses (Supplementary file 1C-G). These results show effect-sizes for DunedinPoAm are larger than for the Horvath, Hannum, and Levine epigenetic clocks. We report comparison to the original Pace of Aging measure only for the Dunedin Study because this original measure could not be computed in any of the other datasets

5) The Abstract is a bit light on numbers – please add the important details such a number of participants, effects sizes, statistical significance estimates for the main results.

We appreciate the reviewer’s comment. However, within the journal’s Abstract word limits, we are not able to add this detail given the large number of analyses in our article.

Reviewer #3:.1) The findings in CALERIE trial are perhaps the most exciting results: mPoA measured at baseline discriminated change in the multivariable composite KDM Biological Age. It is also suggestive of potential to discriminate between treatment arms of a robust intervention on the biology of aging (caloric restriction), though this needs to be followed up with longitudinal assessment. These noteworthy findings should be emphasized. Authors suggested to restructure presentation of results. In the Abstract authors state that mPoA "was disrupted by caloric restriction in the CALERIE trial." mPoA was only measured at baseline not tracked longitudinally for between group effects on change in mPoA as implied by statement. Please revise.

Thank you for this supporting comment. Thank you for the opportunity to clarify. We say “disrupted” because the prediction of future aging made by the DunedinPoAm was disrupted by the intervention: In the control-arm participants, DunedinPoAm correlated with future rate of change in KDM Biological Age. In contrast, in the treatment-arm participants this correlation was disrupted by the intervention. We have revised the Abstract to clarify: “DunedinPoAm prediction was disrupted by caloric restriction in the CALERIE trial”. . Our measure has been renamed “DunedinPoAm” in our revised submission.

2) mPoA measured at single timepoint.• That mPoA is only measured at baseline in CALERIE and at only one timepoint in the validation studies dampens enthusiasm. While the findings in the CALERIE study (Figure 4) are a substantial advancement, biomarkers with repeat measurement are critically needed. Biomarkers with test-retest reliability, sensitivity to change, and predict clinical trial endpoints (e.g. disability onset or incident disease vs. KDM change) are essential to progress. This is noted in limitations. Given iterative nature of research the reviewer trusts that this will be addressed in a future study – hopefully a study repeating mPoA at age 45 in the Dunedin cohort, or at Year 1 in the CALERIE trial when/if additional methylation data become available.

Thank you for these supporting comments. When post-treatment DNA methylation data from CALERIE are available we plan to complete that analysis. But the data are not available now. We note the addition of repeated-measures analysis of change in DunedinPoAm in the Normative Aging Study (Introduction).

• While longitudinal change in mPoA cannot be included in this manuscript, two minor suggestions for context:2A) The TRIIM clinical study should be referenced (a single arm clinical study in 9 men; Fahy et al., 2019). It is currently the only published report of methylation biomarker performance over time in an intervention study.

Done. See Introduction. We note the addition of repeated-measures analysis of change in DunedinPoAm in the Normative Aging Study (Introduction).

2B) The authors correctly state that "measurements of the rate of biological aging are needed to serve as surrogate endpoints in trials…" This is important, and additional context is needed. For example, criteria to validate surrogate endpoints would be pertinent- e.g. Prentice criteria: biomarker to be considered as a surrogate, it should a) be correlated with clinical endpoint; b) capture net effect of intervention on clinical endpoint. This will provide context and forecast next steps in logical progression of biomarker development (e.g. longitudinal assessment and prediction of clinical trial outcome).

Thank you for this suggestion. We have added the suggested text to the final sentence of our penultimate Discussion paragraph.

“Ultimately, to establish DunedinPoAm as a surrogate endpoint for healthspan, it will be necessary to establish not only robust association with healthy lifespan phenotypes and modifiability by intervention, but also the extent to which changes in DunedinPoAm induced by intervention correspond to changes in healthy-lifespan phenotypes (Prentice, 1989).”

3) Dunedin study.• Many of the physical performance tests in Dunedin Study are likely to have ceiling effects and limited change over time would be expected given the ages and health status of the study population. That the investigators find anything useful in change over time to train the methylation clock is a wonder (reviewer anticipated "flat" findings like those for grip strength). Given the history of work with this cohort revision is not requested. In future it would be interesting to see how mPoA would perform if the algorithm were trained in a more diverse population with greater observable deficits in physiologic function over time.

Thank you for these comments. We note that Grip Strength is the only functional test that does not show mean-level decline across the 7-year follow-up interval. As can be seen in the plots, there is substantial variation in rates of decline in this midlife cohort. We can observe this decline because the birth cohort represents the full population, with a retention rate at age 45 over 94%, which means that data for physical function are present for individuals who tend not to take part in typical research samples, including individuals who are experiencing declining function. We agree that follow-up analysis in additional cohorts of older adults will increase knowledge about the measure.

3A) The comparison of adult (age 45) to juvenile (age 7-13) Wechsler Intelligence Scales as an index of 'cognitive decline' is puzzling. Please provide validation and justification.

Thank you for the opportunity to clarify. We compared age-45 Wechsler intelligence-test scores to Wechsler intelligence scores from ages 7-13. The Wechsler test is the most reliable measure of cognitive functioning available, and the most widely used in clinical practice. Comparison of scores on adolescent and adult versions of the test, which are matched for content, distributional properties, and psychometric properties, is a strong method to detect cognitive decline. This analysis comparing midlife IQ to adolescent IQ parallels previously published analyses of cognitive decline in the Dunedin Study (Belsky et al., 2015; Meier et al., 2012; Reuben et al., 2017), as well as in other cohorts (Deary and Batty, 2007).

3B) Please provide references on validation work for the facial aging in the text or supplement.

Done. We have added references to our previous work with this measure (Belsky et al., 2015; Shalev et al., 2014) as well as to the foundational paper on facial aging as a measure of mortality risk (Christensen et al., 2009).

Reviewer #4:A major limitation of previous DNA-methylation based models of aging is the difficulty of arriving at some kind of biological interpretation of their behavior because they are derived from chronological age. mPOA is, by contrast, derived directly from "blood- and organ-system-functional biomarkers". As a result, it should be possible to investigate the relative importance of each of these biomarkers. It could help to explain how and why mPOA differs from POA and where and when it might be more or less informative. At the very least the manuscript could include a table and/or figure comparing the associations of mPOA and POA with the 18 biomarkers.

This is an interesting suggestion. With respect to the question of “relative importance”, the original Pace of Aging measure is a composite of slopes of change across 18 different biomarkers. Each slope is weighted equally in the composite Pace of Aging. As a result, biomarkers with slopes that correlate with slopes of other biomarkers have relatively higher overall correlations with the composite measure. Following the reviewer’s suggestion, we computed correlations of DunedinPoAm with each biomarker slope and compared these to biomarker slope correlations with Pace of Aging. As expected, biomarker slope correlations with DunedinPoAm tend to be smaller as compared to correlations with Pace of Aging. However, the rank orderings of biomarker slope correlations with DunedinPoAm and Pace of Aging are similar, suggesting that biomarker slopes contribute similar information to both measures.

The plot in Author response image 1 graphs biomarker slope correlations with DunedinPoAm (y-axis) against biomarker slope correlations with the original Pace of Aging on which Dunedin PoAm was trained(x-axis). Overall, these data do not suggest any clear differences in organ-system contributions to DunedinPoAm as compared to Pace of Aging. We are happy to take editorial guidance on inclusion of this plot.

There is a distinct lack of negative/null findings in the paper suggesting that such findings may have been omitted. For example, mPOA appears to be associated with all measures of aging (except for grip strength, the only measure that apparently increases with age), and Figure 2—figure supplement 1 suggests that mPOA performs better than POA for all (except grip strength). Some functioning and aging measures used to investigate POA are not included in the present study (see e.g. Belsky et al., 2015). Why are they omitted?

Thank you for the opportunity to clarify. There were three measures studied in our original 2015 paper that were not included in our original submission, retinal venular caliber, retinal arteriolar caliber, and grooved pegboard testing of motor coordination.

We have added the age-45 grooved pegboard data. The results are that faster DunedinPoAm is associated with slower pegboard times at age 45 and with more decline in pegboard speed from 38 to 45, although this latter results is not statistically significant at the α=0.05 level. See Figure 2 and Supplementary file 1C.

Unfortunately, we do not yet have age-45 retinal measures available to include in this paper. The retinal vessel measures are generated from digital retina images by our collaborators at the National University of Singapore. They are currently working on this coding.

If mPOA really is better than POA, this should at least draw some discussion if not some follow-up analysis. It suggests that DNA methylation captures some informative biological variation lacking in the 18 underlying biomarkers, perhaps cell count variation, smoking behavior, or age?

We now comment on the differential performance of DunedinPoAm and the original Pace of Aging measure in the third paragraph of the Discussion. It is possible that Dunedin PoAm captures some information lacking in the 18 biomarkers, but also possible that it contains less noise.

Given the significant amount of published work related to the 'epigenetic clocks', it is critical to show that any new DNA methylation-based model of aging is not only different from previous models but superior in important ways. The manuscript only compares mPOA to other models in terms of associations with KDM Biological Age. In this comparison, mPOA is clearly superior to Horvath and Hannum, but for Levine the comparison is not so clear. It should be easy to derive Levine in the Dunedin data and test associations with functional decline and subjective aging (in fact, haven't Horvath and Hannum have already been derived in Dunedin? – see Belsky et al., 2018).

Done. The substance of the advance represented by the DunedinPoAm is that it is an entirely different kind of aging measure as compared with the so-called clocks. We now explain this in more detail in the Introduction. That said, DunedinPoAm effect-sizes are larger as compared to the epigenetic clocks in all analyses (Supplementary file 1C-G and Figure 1—figure supplements 1 and 3, Figure 2—figure supplements 1-3).

Smoking status appears central to mPOA – of the 46 CpG sites on which mPOA is based, at least 12 have been previously linked with smoking. This is explored only for the comparison with the 'epigenetic clocks' in Understanding Society and for assessing associations with SES and victimization in E-Risk. To what extent does smoking status influence the association between mPOA and POA? To what extent does smoking status influence associations/predictions of functional decline and subjective aging in Dunedin?

Thank you for pointing this out. Dunedin PoAm is not simply a marker of tobacco smoking. We have added results to Supplementary file 1C-E reporting results for analysis with covariate adjustment for smoking in all of the cohorts studied except CALERIE. (CALERIE Trial participants were all non-smokers.) Covariate adjustment for smoking does not change any of our findings, although effect-sizes in some analyses are partly attenuated in smoking-adjusted analysis. The association of the longitudinal 26-38 Pace of Aging with our age-38 methylation measure (now named “DunedinPoAm”) does not vary by smoking status. Figure 1—figure supplement 3.

References:

Christensen, K., Thinggaard, M., McGue, M., Rexbye, H., Hjelmborg, J. v B., Aviv, A., Gunn, D., van der Ouderaa, F., and Vaupel, J.W. (2009). Perceived age as clinically useful biomarker of ageing: cohort study. BMJ 339.

Deary, I.J., and Batty, G.D. (2007). Cognitive epidemiology. Journal of Epidemiology and Community Health 61, 378–84.

Meier, M.H., Caspi, A., Ambler, A., Harrington, H., Houts, R., Keefe, R.S.E., McDonald, K., Ward, A., Poulton, R., and Moffitt, T.E. (2012). Persistent cannabis users show neuropsychological decline from childhood to midlife. PNAS 109, E2657–E2664.

Reuben, A., Caspi, A., Belsky, D.W., Broadbent, J., Harrington, H., Sugden, K., Houts, R.M., Ramrakha, S., Poulton, R., and Moffitt, T.E. (2017). Association of childhood blood lead levels with cognitive function and socioeconomic status at age 38 years and with IQ change and socioeconomic mobility between childhood and adulthood. Jama 317, 1244–1251.

Shalev, I., Caspi, A., Ambler, A., Belsky, D.W., Chapple, S., Cohen, H.J., Israel, S., Poulton, R., Ramrakha, S., Rivera, C.D., et al. (2014). Perinatal complications and aging indicators by midlife. Pediatrics 134, e1315-1323.